

# Circum-Indian ocean hydroclimate at the mid to late Holocene transition: The Double Drought hypothesis and consequences for the Harappan

Nick Scroxton[1,2,3], Stephen J. Burns[1], David McGee[2], Laurie R. Godfrey[4], Lovasoa Ranivoharimanana[5], Peterson Faina[5]

[1]Department of Geosciences, 611 North Pleasant Street, University of Massachusetts Amherst, MA 01030, USA
[2]Department of Earth, Atmospheric and Planetary Sciences, Massachusetts Institute of Technology, 77 Massachusetts Avenue, Cambridge, MA 02139, USA
[3]School of Earth Sciences, University College Dublin, Bellfield, Dublin 4, Ireland
[4]Department of Anthropology, 240 Hicks Way, University of Massachusetts, Amherst, MA 01003, USA
[5]Mention Bassins sédimentaires, Evolution, Conservation (BEC) – BP 906 – Faculté des Sciences, Université d'Antananarivo – 101 Antananarivo, Madagascar

Correspondence to: Nick Scroxton (*nick.scroxton@ucd.ie*)

## Abstract

The decline of the Mature Harappan period of the Harappan civilization in and around the Indus Valley between 4.3 and 3.9 kyr BP, its transition to the Late Harappan and subsequent abandonment by 3.0 kyr BP are frequently attributed to a reduction in summer monsoon rainfall associated with the 4.2 kyr event (4.26-3.97 kyr BP). Yet while the 4.2 kyr event is well documented in the Mediterranean and Middle East, its global footprint is undetermined, and its impact on monsoon rainfall largely unexplored. In this study we investigate the spatial and temporal variability of the tropical circum-Indian ocean hydroclimate in the mid to late Holocene. We conducted Monte-Carlo principal component analysis, taking into account full age uncertainty, on ten high-resolution, precisely dated paleohydroclimate records from the circum-Indian Ocean basin, all growing continuously or almost continuously between 5 and 3kyr BP. The results indicate the dominant mode of variability in the region was a drying between 3.967 kyr BP (±0.095 kyr standard error) and 3.712 kyr BP (±0.092 kyr standard error) with dry conditions lasting for at least 300 years in some records, but a permanent change in others. We interpret PC1 and the drying event as a proxy of summer monsoon variability. A more abrupt event from 4.2 to 3.9 kyr BP is seen locally in individual records, but is often not of unusual magnitude, lacks regional coherence and is of minor importance to the principal component analysis. This result does not fit the prevailing narrative of a summer monsoon drought at the 4.2 kyr event contributing to the decline of Harappan civilisation. Instead we present the "Double Drought Hypothesis". A comparison of existing Indian subcontinent paleoclimate records, modern climatology, the spatial and temporal evolution of Harappan archaeological sites, and upstream climatic variability in the Indian Ocean and Mediterranean and Middle East indicates two consecutive droughts were contributing factors in the decline of the Harappan. The first drought was an abrupt 300-year long winter rainfall drought between 4.26 and 3.97 kyr BP, associated with the 4.2 kyr event, propagated from the Mediterranean and Middle East. This led to Harappan site abandonment in the Indus valley and the end of Mature Harappan period. The second drought was a more gradual but longer lasting reduction in summer monsoon rainfall beginning 3.97 kyr BP leading to the further site abandonment





at sites in Gujarat, a transition towards a more rural society, and the end of the Late Harappan. The consequences for the new mid to late Holocene Global Boundary Stratotype Section and Point in a stalagmite from Meghalaya are explored.

## 1 Introduction

The Holocene (the last 11,700 years) is being increasingly recognized as a period of significant climate variability (Fleitmann et al., 2003; Gupta et al., 2003; Mayewski et al., 2004). Understanding the patterns and processes of recent variability is important in understanding the modern climate system's response to perturbation. In the Northern Hemisphere mid-latitudes the 4.2 kyr event (4.26-3.97 kyr BP (Carolin et al., 2019)) is one of the largest climate anomalies of the Holocene (Bini et al., 2019; Booth et al., 2005; Kaniewski et al., 2018), representing either a relatively modest forced event such as a freshwater input to the North Atlantic (Wang et al., 2013), or a very large unforced event (Yan and Liu, 2019). As one of the largest perturbations during the Holocene, the 4.2 kyr event was recently designated as the chronostratigraphic boundary between the Northgrippian and Meghalayan subdivisions of the Holocene epoch (Walker et al., 2018).

Because of chronological uncertainties inherent in paleoclimate records, the areal extent of the 4.2 kyr BP event beyond the data-rich heartland of Mediterranean Europe (Bini et al., 2019) and Mesopotamia (Kaniewski et al., 2018) is unclear. In the tropics and monsoonal areas, climate anomalies with age uncertainties that overlap 4.26-3.97 kyr BP are often attributed to the 4.2 kyr event (Marchant and Hooghiemstra, 2004; Staubwasser and Weiss, 2006). This includes dry conditions in tropical Africa (Marchant and Hooghiemstra, 2004), wet conditions in tropical South America (Marchant and Hooghiemstra, 2004), changes to ENSO (Toth and Aronson, 2019) and aridification in India (Dixit et al., 2014; Dixit et al., 2018). However, these changes are often significantly longer than the <300-year climatic excursion of the 4.2 kyr event and/or date to slightly later. The global extent of the 4.2 kyr event is uncertain.

The 4.2 kyr event also garners attention because of its synchronicity to major societal transformations (Höflmayer, 2017; Wang et al., 2016), providing test cases for exploring the role of climate in societal change (Butzer and Endfield, 2012; deMenocal, 2001; Weiss and Bradley, 2001). For example, the end of the Akkadian civilization in northern Mesopotamia at 4.2 kyr BP (Weiss et al., 1993; Weiss, 1997) can be related to contemporaneous drying through the presence and provenance of a well-dated 300-year increase in regional dustiness (Carolin et al., 2019; Cullen et al., 2000).

Similar to climatic and environmental changes, societal changes in civilizations outside of Mediterranean Europe and Mesopotamia with age error uncertainties that overlap the 4.26-3.97 kyr BP are also often attributed to the 4.2 kyr event. Significant attention has been focused on the societal changes and deurbanization at the end of the Mature Harappan period in and around the Indus valley around 3.9 kyr BP (Dixit et al., 2014; Petrie et al., 2017). The Mature Harappan describes an





approximately 700-year long period from 4500-3800 yr BP. During this time a dense distribution of over 700 settlements, including four or five cities in Gujarata and along the Indus and Ghaggar-Hakra rivers, underwent a standardization of material culture and urban planning ((Petrie et al., 2017) and references within). The decline of Harappan civilization sees a deurbanization and migration of sites towards the headwaters of the Ghaggar-Hakra, hypothesized to reflect habitat tracking (Gangal et al., 2010). Given errors on radiocarbon ages, the timing of the decline is uncertain, beginning as early as 4300 yr BP (Sengupta et al., 2020), likely well underway by 3900 yr BP (Giosan et al., 2012). The Harappan civilization continued in a more limited form well beyond the 4.2kyr event, known as the Late Harappan. However, by 3000 yr BP, all major urban sites had been abandoned, and the Harappan had become a more fragmented rural society, based in southern Gujarat and the Ganga-Yamuna interfluve.

An association of the end of the Mature Harappan period and the 4.2 kyr event has been widely made in the literature (Berkelhammer et al., 2012; Dixit et al., 2014; Dixit et al., 2018; Giosan et al., 2012; Kathayat et al., 2017; Staubwasser et al., 2003). The assumed mechanism is usually a decline in summer monsoonal rainfall. However, the idea has been resisted by some in the archaeological community for several reasons: 1) There is evidence for long-term drying over the course of the Holocene, suggesting that the Mature Harappan developed in response to water stress rather than pluvial conditions (Gupta et al., 2003; Madella and Fuller, 2006). 2) There is evidence for drying, adaptation and continuation within the Mature Harappan (Pokharia et al., 2017). There are also two key climate problems with this interpretation. 1) evidence for drying in the subcontinent does not typically support an abrupt 300-year long drought with subsequent return to wet conditions, as might be expected if the 4.2 kyr event were the cause (Madella and Fuller, 2006; Wright et al., 2008). Second, the most likely climate mechanism of change to the Indian Summer Monsoon, a shift in ENSO state and increased El Niño frequency, occurs 300 years too late (MacDonald, 2011). While the causes of societal decline and transitions are undoubtably multifaceted and complex, the role of climate in the end of the Mature Harappan requires further study.

To investigate key questions concerning 1) the areal extent of the 4.2 kyr event, and 2) how climate impacted the Harappan civilization, high resolution paleoclimate records are needed. With age errors often less than 1% (Cheng et al., 2013), speleothems provide high resolution, precisely dated records of past hydroclimate changes capable of providing insight into sub-centennial scale Holocene climate variability (Bar-Matthews and Ayalon, 2011; Fleitmann et al., 2003). Recent publication of new speleothem records in the last three years from west and east India (Kathayat et al., 2017; Kathayat et al., 2018), Sumatra (Wurtzel et al., 2018), Rodrigues Island (Li et al., 2018), and Madagascar (Wang et al., 2019)(Scroxton et al., counterpart paper), along with a high resolution sediment core from the Arabian Sea (Giesche et al., 2019), have dramatically increased the number of high-resolution paleoclimate records of the circum-Indian Ocean basin hydroclimate. The available records now cover much of the circum-Indian Ocean basin, including the west, central and east Indian Ocean, both northern





and southern hemispheres, and equatorial latitudes. The new data now allow for the statistical analysis of changes in hydroclimate and assessment of regional versus local changes.


In this study we investigate the spatial patterns and reach of climate variability around the Indian Ocean (hereafter circum-Indian Ocean basin) during the mid to late Holocene. We take advantage of the abundance of new paleoclimate records to investigate regional coherency (or lack of) in circum-Indian Ocean basin hydroclimate by conducting Monte-Carlo principal component analysis (MC-PCA)(Deininger et al., 2017) on a suite of high resolution, precisely dated regional hydroclimate

records (Figure 1, 2, Supplementary Table 1). The speleothem records used in the MC-PCA are mostly interpreted as proxies for summer monsoon rainfall changes through regional circulation changes, regional and local rainfall amount (Supplementary Discussion 1), and have near continual coverage from 5000 to 3000 yrs BP. In addition, we also analyze three paleoclimate records from two marine sediment cores with annual laminations and proxy records interpreted as responding to terrestrial hydroclimate variability. Together these records allow us to investigate the impacts of the 4.2 kyr event on tropical Indian

Ocean basin monsoonal rainfall, within the context of the middle to late Holocene.

## 2 Methods

We conducted Monte-Carlo principal component analysis (MC-PCA) on high-resolution records from the circum-Indian Ocean basin using the technique and MATLAB code from (Deininger et al., 2017). In summary: For each record, 2000 age models are created by allowing ages to vary within 1 standard deviation assuming a Gaussian distribution and then linearly

interpolating between the records (Supplementary Figure 1). This creates age models with different temporal resolutions based on the sample spacing. The age models are upscaled to 15 (PCA-1, PCA-2) or 20 (PCA-3) year-long bins using a Gaussian kernel applied to all data points within the bin. Climate records are normalized to mean 0 and standard deviation 1. Principal component analysis (PCA) is then conducted to the 2000 sets of upscaled normalized proxy records. Age uncertainties can result in bimodal distributions of principal component (PC) time series. A flipping procedure is used to reinvert the flipped

PCs. The final PCA time series is calculated using the mean and standard deviation for each bin. The Kolmogorov-Smirnov (KS) test is used to ensure that only statistically meaningful (95% level) principal components are analyzed.

PCA requires continuous, regularly spaced data. Two of the records: the KNI composite record from Australia (Denniston et al., 2013) and the AK1 record from Madagascar (Scroxton et al., counterpart submission) contain short hiatuses. The hiatus in

the Madagascar record is replicable across two different speleothems ((Wang et al., 2019) Scroxton et al., counterpart submission) from two different caves and likely represents dry conditions rather than a drip specific or cave specific changes in ventilation. Similarly, the KNI record from Australia is composed of multiple stalagmites. The gaps between stalagmites can be treated as hiatuses in the composite record as the age models contain gaps. However, the age uncertainties are such that



the records could be considered continuous within error. We chose to treat the gaps as hiatuses. We filled in the hiatuses with
dummy $\delta^{18}O$ values corresponding to dry conditions using $\delta^{18}O$ values 1.5 standard deviations above the mean $\delta^{18}O$ for each
record between 5 and 3 kyr BP (Supplementary Figure 1). 1.5 standard deviations ensured that the dummy values were more
positive than the stable isotopes values before and after the hiatuses (i.e. dry conditions) but did not assume that hiatuses
represented the most positive values on the record (driest conditions). This is because speleothems drying out is likely a
function of the length of drought as it is the severity, and as $\delta^{18}O$ is not necessarily a direct measure of rainfall amount it would
be incorrect to assume a lack of growth represented the most positive $\delta^{18}O$ values for a continuous period of time. To test
whether the dummy data had a significant effect on the MC-PCA, we conducted the analysis both with (PCA-2) and without
(PCA-1) the Australian and Malagasy speleothem records. We found little difference between the two, suggesting the dummy
data has little to no effect on the shape or timing of the variance explained by the first two principal components (Figure 3).

For the purposes of the MC-PCA code we also assumed that the two offshore Indus records (Giesche et al., 2019) had
independent age models. As the two records are from the same core this is not true. This assumption introduces a small error
to the MC-PCA, slightly under-estimating the similarity between the two records and over-estimating the confidence interval
bounds in the principal component time series.

To detect the changepoints in PC1 we used RAMPFIT (Mudelsee, 2000; Mudelsee, 2013), a Fortran program that estimates
change-points using a weighted least squares regression for the proxy value, a brute force search for the time value, and a
moving block bootstrap to estimate uncertainty. We constrained the search to find one changepoint between 3.1 and 3.8 kyr
BP and one changepoint between 3.8 and 4.5 kyr BP.

## 3 Results

To assess the commonality that might be present in circum-Indian Ocean basin hydroclimate records, we conducted three
different Monte-Carlo principal component analyses with sets of ten high resolution hydroclimate records from around the
Indian Ocean basin from 5000–3000 years ((Chen et al., 2016; Denniston et al., 2013; Deplazes et al., 2013; Fleitmann et al.,
2007; Giesche et al., 2019; Kathayat et al., 2017; Li et al., 2018; Wurtzel et al., 2018), Scroxton et al., counterpart paper)
(Figure 4, 5). The first PCA (PCA-1) was run on the five continuous speleothem $\delta^{18}O$ records from Oman, Rodrigues, W.
India, Sumatra and Borneo at 15 year resolution. The second PCA (PCA-2) includes the five previous speleothem $\delta^{18}O$ records,
plus two speleothem records from Madagascar and Australia which have short hiatuses filled with dummy $\delta^{18}O$ data. The third
PCA (PCA-3) includes all seven speleothem $\delta^{18}O$ records, plus three laminated marine sediment core records from the Arabian
Sea, with proxies interpreted as responding to terrestrial hydroclimate change (Table S2, Supplementary Discussion S2). PCA-
3 was run at 20-year resolution to accommodate the lower sampling resolution of the marine sediment cores.




All three analyses produce similar results. The first principal component (PC1) is dominated by a gradual shift to dry conditions beginning at 3967 yr BP (±95 yr standard error). The gradual drying indicated by PC1 is a feature of many of the original records and not the result of a regionally asynchronous abrupt transition (Figure 3). Drying associated with PC1 lasts until 3712 yr BP (±92 yr standard error).


The majority of records have negative loadings on the PC1 score (Figure 4). The drying was therefore widespread across most of the Indian Ocean basin. A positive loading (wetting trend) is observed in the Borneo speleothem record. This opposite response of the Indo-Pacific Warm Pool to the more homogenous response in the circum-Indian Ocean basin suggests that zonal climatic processes may dominate over meridional processes in this low frequency variability (Denniston et al., 2013; 
Dutt et al., 2018; Prasad et al., 2014; Toth and Aronson, 2019).

The second principal component is a fluctuating wet and dry signal (Figure 3). For positively loading records (e.g. NW India): wetter conditions occur between 5.0-4.9, 4.5 to 3.7 and 3.2 to 3.1 kyr BP and dryer conditions 4.9 to 4.5 and 3.7 to 3.2 kyr BP (Figure 4). A zonal dipole is possibly also observed in the PC2 of PCA-1 and PCA-2, but this time with the Sumatra record 
plotting with the Borneo record with negative loadings, most records with small loadings and the western India speleothem record with positive loading.

For PCA-3 with the additional sediment cores, PC2 has a different shape than in the other analyses. An additional dry period is detected in PC2 between 4.2 and 3.9 kyr BP. This additional feature changes the PC2 loadings, such that only the sediment 
core records from the Arabian Sea (and Australia) load positively. This result is suggestive of a possible local signal of the 4.2 kyr event.

## 4 Discussion

The absence of a significant, widespread 4.2 kyr event in tropical Indian Ocean hydroclimate has consequences for the timing and causes of the deurbanization of the Harappan civilization in the Indus valley. The Harappan civilization began around 5.2 
kyr BP (Possehl, 2002), reaching its peak between 4.8 and 3.8 kyr BP, during the Mature Harappan period. Cultural decline, including poor pottery craftsmanship and reduced reservoir maintenance or abandoned water reservoirs, began as early as 4.3 kyr BP (Sengupta et al., 2020) leading into the Late Mature Harappan period with societal reorganization, site abandonment and migration underway before 3.9 kyr BP (Possehl, 2002; Sengupta et al., 2020; Vahia and Yadav, 2011). During the Late Harappan period (3.9-3.0 kyr BP), technological decline, settlement size reductions and abandonment continued (Possehl, 
1993; Possehl, 1997; Possehl, 2002; Sengupta et al., 2020). By 3.0 kyr BP, Post-Urban Harappan society had become rural





with few cultural similarities to the Harappan. Post-Urban Harappan settlements were concentrated around the Ganga-Yamuna interfluve and southern Gujarat, which have higher summer rainfall than the Indus Plain (Gangal et al., 2010; Petrie et al., 2017). Here, we review the available archaeological, climatological and paleoclimatological evidence and propose a new hypothesis to explain the decline and abandonment of the Harappan civilization.

**4.1 Spatial and temporal variability in Harappan site abandonment**

There is geographic variation to the timing of abandonment (Figure 5, 6b). At the more northerly sites in the Ghaggar-Hakra and Indus plain, Kot Diji and Rakhighari show no dates beyond 4.25 kyr BP. Kalinbangan has radiocarbon dates as late as 4.0 kyr BP but the region is largely devoid of radiocarbon dates beyond this period. Further south, sites in Gujarat consistently extend beyond 4.0 kyr BP at Prabhas Patan, Rojdi, Nageswar, Vagad, Lothal, Khirsara, Kanmer, Ratanpura and Surkotada. Of

note is the climatological difference between the late abandonment sites in Gujarat and the early abandonment sites in the Indus and Ghaggar-Hakra valleys (Figure. 5, 6d). The two river basins lie in the transition zone between winter dominated rainfall to the north-west and summer dominated rainfall to the south-east. Winter rainfall is largely derived from western disturbances originating in the eastern Mediterranean (Dimri et al., 2015), while summer rainfall is largely sourced from the Arabian Sea via the Indian Summer Monsoon. Therefore, more northerly sites derive a significant proportion of annual rainfall

from winter rainfall, while southern rainfall is almost entirely from the monsoon.

**4.2 Paleoclimate records on the Indian subcontinent**

Paleoclimate evidence for drying on the subcontinent between 5 and 3 kyr BP falls into three, non-mutually-exclusive temporal patterns. The first climatic change is a long-term unidirectional drying over the entire 5 to 3 kyr period. For example: a gradual change in temperature and rainfall between 4.8 and 3.0 kyr BP is observed at PT Tso Lake (Mehrotra et al., 2018).


The second climatic change matches the timing and style of the 4.2 kyr event in the Mediterranean, i.e. a short 300-year dry period with return to wet conditions. Paleoclimate records showing this pattern include high resolution records such as the foraminiferal *Globigerinoides ruber* $\delta^{18}$O record (Giesche et al., 2019; Staubwasser et al., 2003) from the Indus fan, and the KM-A speleothem record from Mawmluh Cave (Berkelhammer et al., 2012). Lower resolution records include lithology,

pollen, biomarkers and trace elements from Lonar Lake which suggest a dry period from 4.6 to 3.9 kyr BP, peaking at 4.2 to 4.0 kyr BP, with wet conditions continuing until 3.7 kyr BP (Prasad et al., 2014).

The third climatic change is a 4.0 to 3.5 kyr BP gradual drying. It is typically associated with wet conditions between 4.5 and 4.0 kyr BP and sometimes a partial recovery from 3.5 to 3.0 kyr BP. This third pattern is observed in the Sahiya Cave

speleothem $\delta^{18}$O record (Kathayat et al., 2017), the *Neogloboquadrina dutertrei* $\delta^{18}$O record from the Indus fan (Giesche et



al., 2019), and the speleothem $\delta^{18}$O records of stalagmites ML.1 and ML.2 from Mawmluh cave (Kathayat et al., 2018). For a full discussion of the Mawmluh cave records see section 4.5.

Expressions of the third pattern in lower resolution or short paleoclimate records include abrupt drying at paleolake Kotla
Dahar at 4.1 kyr BP as recorded in ostracod $\delta^{18}$O, with no recovery (Dixit et al., 2014). Wet conditions at Surinsar Lake occurred between 4.4 and 4.0 kyr BP (Trivedi and Chauhan, 2009). While likely also influenced by westerlies, the Puruogangri ice cap is interpreted as a tropical record (Thompson et al., 2006). Ice cap $\delta^{18}$O shows a decrease starting at 4.0 kyr BP with partial recovery at 3.4 kyr BP. The Didwana salt lake became ephemeral around 4.0 kyr BP (Singh et al., 1990). Gradual drying is recorded in peat bog pollen from Garhwal between 4.0 and 3.5 (Phadtare, 2000), and in speleothem $\delta^{18}$O from Sainji Cave
between 4.0 and 3.5 kyr BP (Kotlia et al., 2015). The paleolake Karsandi record shows wet conditions from 5.1 to 4.5 kyr BP, followed by drying (Dixit et al., 2018). The length of drying is unknown due to an undated massive gypsum layer, but there is partial recovery prior to 3.2 kyr BP. A stalagmite from Dharamjali Cave in the Himalaya shows a gradual drying trend between 3.7 and 3.2 kyr BP (Kotlia et al., 2018). Similar changes are also observed in peninsular India, with a period of wet conditions followed by dry conditions between 4.5 and 3.3 kyr BP observed in the lake pollen record from Shantisagara Lake (Sandeep
et al., 2017).

The three groups are not mutually exclusive. The Lake Rara Mn/Ti record shows a weak monsoon starting early at 4.7 and a partial recovery at 3.5 kyr BP (Nakamura et al., 2016). Similarly, the Lake Agung Co carbonate $\delta^{18}$O shows a gradual drying trend, but with termination of deposition at 3.9 kyr BP (Morrill et al., 2006). Meanwhile the Al/Ca record of Tso Moriri shows
wetter conditions from 5.32 to 4.35 kyr BP, followed by a dry excursion that lasts until 3.4 kyr BP (Dutt et al., 2018). The multiproxy Tso Kar lake record shows a unidirectional drying transition starting around 4.8 kyr BP, with driest conditions around 4.2 kyr BP (Wünnemann et al., 2010).

All of these paleoclimate records are proxies for parts of the hydroclimate system. However, they cannot all be recording
regional variability in the summer monsoon. Instead different proxy systems might record local effects, have variable system memory, be affected by changing temperatures, evaporation rates and cloud cover differently, or show sensitivity to different seasons. To elucidate which pattern might correspond to which season or climatic control, we compare the three Indian subcontinent hydroclimate patterns with wider regional hydroclimate.

### 4.3 Regional climate drivers

The gradual drying seen in some Indian subcontinent paleoclimate records between 5 and 3 kyr BP is likely part of the gradual reduction in northern hemisphere tropical and subtropical rainfall over the course of the Holocene (Haug et al., 2001; Schneider et al., 2014). This underlying secular trend in Holocene climate is caused by decreasing summer insolation from changing





orbital precession. In the Indian Summer Monsoon domain this trend is clearly expressed in full Holocene length speleothem
records from Oman (Fleitmann et al., 2003) and Mawmluh Cave (Berkelhammer et al., 2012). A gradual trend in forcing can

lead to threshold changes in climate due to feedbacks in the climate system, e.g. the termination of the African Humid Period
(Tierney and deMenocal, 2013). Therefore, unidirectional dying events on the Indian subcontinent throughout the Holocene
may be due to threshold effects and local feedbacks on this secular trend.

The second climate signal in the Indian subcontinent is an abrupt drying between 4.2 and 3.9 kyr BP with a return to wet

conditions afterwards. This signal matches the expression of the 4.2 kyr event in the Mediterranean and Middle East as a major
drought (Figure 6c). Western disturbances bring winter rainfall from the Mediterranean across the Middle East to the Indus
valley, controlling both winter rainfall, and spring river flow via winter snowfall in the Himalayas (Dimri et al., 2015). Carbon
isotope values of rice grains, a summer crop, show increased drought only after 4.0 kyr BP in both Gujarat and Indus valley
locations, indicative of no substantial change to summer rainfall between 4.2 and 3.9 kyr BP (Kaushal et al., 2019). The 4.2

kyr event is likely to be a winter rainfall drought, with weakened western disturbances originating in the Mediterranean reduced
winter rainfall in the Indus valley (Dimri et al., 2015; Syed et al., 2006). However, the exact climate driver is unclear. While a
negative excursion of the North Atlantic Oscillation close to 4.2 kyr BP (Olsen et al., 2012) would cause a reduction in rainfall
from western disturbances (Syed et al., 2006), the pattern of Mediterranean suggests a more complex set of drivers than a
simple North Atlantic Oscillation excursion influencing winter rainfall (Bini et al., 2019; Kaniewski et al., 2018). Regardless

of the climate driver, we interpret the 4.2-3.9 kyr BP drying in the Indus valley as a winter rainfall drought.

The third climate signal in the Indian subcontinent is a drying trend from 4.0 to 3.5 kyr BP. The signal matches the PC1 of our
circum-Indian Ocean basin hydroclimate PCA (Figure 6a) and is the dominant hydroclimate variability in the tropical Indian
Ocean at the time. Our PCA focuses only on the highest resolution, most precisely dated records available, and is therefore

dominated by speleothem records. Speleothem records tend to record amounted-weighted changes in precipitation $\delta^{18}O$. As a
result, they are likely biased towards local summer monsoonal rainfall. The 4.0 to 3.5 kyr BP drying trend can therefore be
interpreted as the dominant regional signal of monsoon rainfall. In the Indus valley, this would be the Indian Summer Monsoon.
The potential summer bias in the PCA might also explain why a winter dominated 4.2 kyr event is not strongly expressed in
any of the PCs in our analyses.


The interpretation of the 4.0 to 3.5 kyr BP drying trend as a tropical climate signal matches numerous other tropical
hydroclimate records. The inferred mechanism is an abrupt change in ENSO behavior around 4.0 kyr BP with an increase in
the frequency of El Niño events, and/or a narrowing of the topical rainbelt (de Boer et al., 2014; Denniston et al., 2013; Gagan
et al., 2004; Giosan et al., 2018; Li et al., 2018; MacDonald, 2011; Toth et al., 2012). In the modern climate system, El Niño





events are associated with reduced Indian summer monsoon precipitation (Mooley and Parthasarathy, 1983). Therefore, an increase in El Niño events may have caused a decrease in time-averaged precipitation in the Indus valley.

### 4.4 The Double Drought hypothesis

We propose the "Double Drought hypothesis" to explain the decline of the Mature Harappan and abandonment of the Late
Harappan. The hypothesis combines geographic variation in the timing of abandonment (Figure 6b) and differences in local rainfall seasonality (Figure 5, 6d) with an understanding of 'upstream' hydroclimate changes in the Indian Ocean basin (Figure 6a) and Mediterranean (Figure 6c) which influence summer and winter rainfall respectively.

An abrupt decline in winter rainfall from 4.26 to 3.97 was immediately followed by a more gradual but larger and longer
lasting decline in summer rainfall. Drying occurred between 3.97 and 3.71 kyr BP, and lasted for several centuries, in some areas being a permanent change in rainfall. Together these two droughts caused climate stress to the Harappan civilization and their trading partners. The winter rainfall drought is associated with the end of the Mature Harappan and abandonment of more northerly settlements. The summer rainfall drought is associated with the gradual abandonment of all remaining major Harappan settlements during the Late Harappan and transition to more rural society. Adaptation to changing climate is clear
from the archaeological evidence. Therefore, the style, timing and location of societal response is determined by the complex interaction of societal, biogeophysical and geomorphological feedbacks and responses, rather than by climate alone.

Both droughts occur on an underlying secular trend of decreasing northern hemisphere tropical summer rainfall over the course of the Holocene, driven by decreasing insolation. The Harappan civilization grew, thrived and declined on this gradual trend,
and the archaeological evidence points to a mastery of water resources. However, this first-order trend suggests that a drought driven decline in the civilization was a matter of when not if.

Superimposed on this multi-millennial scale trend is higher-frequency centennial-scale variability which would have led to periods of relatively wet and dry conditions. The Mature Harappan period likely coincided with one of these wetter periods
(Dixit et al., 2018; Giosan et al., 2018; Singh, 1971; Wright et al., 2008). The decline of the Mature Harappan and abandonment of the Late Harappan therefore occurred during two consecutive multi-centennial scale droughts superimposed on first order millennial scale drying.

The winter rainfall drought between 4.26 and 3.97 kyr BP was likely caused by a reduction in western disturbances that bring
rainfall from the Mediterranean and Middle East (Figure 6c). Under modern climate the proportion of winter rainfall varies greatly across the area of Harappan occupation, from couple of percent in Gujarat, through to 20% at the Indus valley and





Ghaggar-Hakra sites, to greater than 90% on the western boundary of the Harappan civilization in western Pakistan (Figure 5, 6d)(MacDonald, 2011). During the Mature Harappan prior to the drought, enhanced winter rainfall may have played an even larger role in annual rainfall (Giosan et al., 2018). Therefore, a significant reduction in winter rainfall could still cause

significant climate stress.

The societal response to the climate stress caused by winter rainfall drought was significant, as evidenced by reduced maintenance, poor pottery craftsmanship, abandoned water reservoirs (Sengupta et al., 2020) and the abandonment of settlements in areas with a higher proportion of winter rainfall such as the Indus valley (Figure 6b). Trading relationships with

civilizations in the Middle East heavily reliant on winter rainfall would have also likely been hit (MacDonald, 2011).

However, the Harappan were able to adapt. The Harappan grew both winter and summer rainfall crops and during the Mature Harappan switched from barley and wheat to millet (Petrie et al., 2017; Petrie and Bates, 2017; Pokharia et al., 2017). At present, barley and wheat are winter crops and millet a summer crop (Petrie and Bates, 2017). The agricultural evidence

therefore suggests a switch from winter to summer crops. There are some important caveats and nuance to such a simple interpretation though, namely 1) there is substantial geographic variation, 2) winter crops are reliant on residual soil moisture from the summer rains, 3) the Harappan may have engaged in multicropping or strategies not aligned with modern practices and 4) the timing of rainfall and inundation may not be simultaneous owing to snow melt (Petrie and Bates, 2017). Regardless, agriculture adaptation strategies and continued summer rainfall ensured that the change was not critical to the Harappan

civilization, which continued, albeit in a reduced form, beyond 3.97 kyr BP and the climatic recovery from the winter drought.

At a similar time to the winter rainfall recovery at 3.97 kyr BP, a multi-centennial decline in summer monsoonal rainfall began that was both more severe and substantially longer than the 4.2 kyr event (Figure 3, 6a). This second drought was the response to a shift in the mean state of tropical zonal circulation, and possibly an increase in El Niño frequency. We argue that it was

this second transition that played the final role in the Late Harappan deurbanization over the course of the following centuries. The transition to Post-Urban Harappan societies saw migration to two areas: the Ganga-Yamuna interfluve and southern Gujarat, areas with higher summer rainfall amounts.

The decline in summer rainfall was relatively gradual, occurring over 250 years rather than the decades of the 4.2 kyr event,

and lasting for hundreds of years more. Further, abandonment of Late Harappan sites was not coincident and we know that the Harappan were adaptable and indeed thrived in the face of hydrological variability. Therefore, the nature, location, and precise timing of societal change in response to the two consecutive droughts are likely due to complex biogeophysical (Cookson et al., 2019), geomorphological (Giosan et al., 2012) and/or social processes and feedbacks (Madella and Fuller, 2006; Petrie et al., 2017; Schug et al., 2013).






The Double Drought hypothesis is consistent with the timing, location, and long-lasting nature of societal change during the decline and abandonment of the Harappan civilization. While this hypothesis is consistent with many paleoclimate records from the Indian subcontinent, it is not always consistent with their prior interpretation. For example, our hypothesis is in direct contrast to that of Giesche et al. (2019) who used foraminifera $\delta^{18}O$ records (interpreted as Arabian Sea salinity changes) to

hypothesize a temporary reduction in summer rainfall at 4.2 kyr BP, and a longer lasting reduction in winter rainfall at 3.9 kyr BP. To reconcile these ideas, paleohydroclimate proxies with less ambiguous seasonality than speleothems and foraminifera will be required.

Our Double Drought Hypothesis is consistent with broader climatic changes in 'upstream' areas of Indus valley rainfall;

changes in winter mid-latitude westerlies are interpreted as influencing winter rainfall, while changes in tropical Indian Ocean hydroclimate are interpreted as influencing summer monsoonal rainfall.

### 4.5 Consequences for the mid to late Holocene GSSP

The Mawmluh Cave speleothem records which define the 4.2 kyr event (Berkelhammer et al., 2012; Walker et al., 2018) are too short to be included in our PCA analysis. However, the highest resolution replicated record from the cave, ML.1 (Kathayat

et al., 2018), shows wetter than normal conditions at 4.1–4.0 kyr BP followed by longer term multi-centennial drying – consistent with PC1. The Global Boundary Stratotype Section and Point (GSSP) golden spike is located in stalagmite KM-A (Berkelhammer et al., 2012). Stalagmite KM-A contains two increases in $\delta^{18}O$ (drying events), one at 4.31 kyr BP and one at 4.05 kyr BP. While the timing of these changes might be consistent with drying anomalies for both a 4.26 and a 3.97 kyr BP drying event, there are notable issues with the stalagmite itself. The KM-A record replicates neither the other speleothem from

Mawmluh Cave (Kathayat et al., 2018) nor any regional records (this study). While the ages of KM-A are very precise, there are only three ages between 5084 and 3654 yr BP and stalagmite growth is very slow. The shape of the stalagmite after 5 kyr BP is not convincing of unaltered equilibrium deposition. Even if the $\delta^{18}O$ record of KM-A does record both drying events, the golden spike location is defined as 4.20 kyr BP, part of a run of 40 consecutive $\delta^{18}O$ samples between the two droughts, with relatively minor variability (<1‰). The existence of the Northgrippian-Meghalayan GSSP golden spike in a low

resolution, low dating frequency record, not replicable within its own locality, ambiguously defining a climate event (or two) that is not significant across its climatic domain, is problematic at a minimum (Helama and Oinonen, 2019). We recommend that an alternative GSSP golden spike for the mid- to late-Holocene transition be identified.





## 5 Conclusions

Monte-Carlo principal component analysis of ten high-resolution paleoclimate records from the region show tropical
hydroclimate variability between 5 and 3 kyr BP in the circum-Indian ocean basin is dominated by a region-wide drying
beginning at 3.97 kyr BP. The 4.2 kyr event had limited impact on tropical monsoon precipitation.

We combine the evidence from Harappan radiocarbon dates, analysis of paleoclimate records from across the Indian
subcontinent, modern subcontinent climatology, and upstream paleoclimate records from the Mediterranean and circum-Indian
Ocean basin to create the "Double Drought Hypothesis" to explain the decline and abandonment of the Harappan civilization.
On a millennial scale secular drying trend, two back to back centennial scale droughts combined to cause significant and
eventually critical climate stress to Harappan society. The first drought between 4.26 and 3.97 kyr BP, associated with the 4.2
kyr event, was caused by a reduction in winter rainfall. Site abandonment occurred in the Indus and Ghaggar-Hakra valleys,
but in Gujarat, there was adaptation towards summer crops. The first drought resulted in the end of the Mature Harappan
period. The second drought started at 3.97 kyr BP and saw 250 of drying followed by centuries of drought, if not a permanent
shift in rainfall. This second drought was caused by a reduction in summer monsoon rainfall due to a global shift in the mean
state of the tropics, possibly via an increase in El Niño event frequency. This second drought led to the abandonment of
remaining Harappan sites, the end of the Late Harappan, and transition to a more rural society in southern Gujarat and the
Ganga-Yamuna Interfluve. The double droughts were a significant factor in the end of the Harappan civilization, causing
significant climate stress to Harappan society. Other biogeophysical and geomorphological changes in response to changing
rainfall patterns likely also contributed to societal stress. Ultimately, social responses to these pressures controlled the nature,
timing and location of adaptation, abandonment and migration of Harappan society.

The findings presented here have implications for the attribution of the 4.2 kyr event as a global climatic event. While some
individual records from the circum-Indian Ocean basin show locally significant climate excursions contemporaneous with the
4.2 kyr BP event, when viewed basin wide the event has little regional coherence nor is it of unusual severity at most sites.
The 4.2 kyr event therefore had limited impact on tropical monsoonal rainfall around the circum-Indian Ocean basin, Previous
studies may have falsely attributed low latitude, low frequency climate variability to the 4.2 kyr event when there are other
significant changes in the tropical climate system at similar times. We propose that the 4.2 kyr event had a more limited
geographical impact than is commonly assumed.

## Data availability

Data are available from the authors (nick.scroxton@ucd.ie) and at the NOAA Paleoclimatology Database:
https://www.ncdc.noaa.gov/paleo/study/xxxxx.



**Author Contributions**

NS conceived the study, ran data analysis, and was primarily responsible for writing the manuscript. SJB, DM and LRG provided guidance on project direction, discussion of findings and helped write and edit the manuscript. LR and PF contributed to the development and interpretation of the records from Madagascar.

**Competing interests**

The authors declare no competing interests.

**Acknowledgements**

NS, SJB and DM acknowledge support from NSF award AGS-1702891/1702691, LRG and SJB from NSF award BCS-1750598 and DM from NSF award EAR-1439559 and the MIT Ferry Fund. Fieldwork in northwest Madagascar was conducted under a collaborative accord for paleobiological research between the University of Antananarivo (Département de Paléontologie et d'Anthropologie Biologique) and the University of Massachusetts (Department of Anthropology); collaborative work was further supported under a second accord for paleobiological and paleoclimatological research between the University of Antananarivo (Mention Bassins sédimentaires, Evolution, Conservation) and the University of Massachusetts Amherst (Departments of Anthropology and Geosciences). The research was sanctioned by the Madagascar Ministry of Mines, the Ministry of Education, and the Ministry of Arts and Culture.






**Figure 1: Location map. Squares denote locations of stalagmites; circles denote marine sediment cores. Closed symbols are used in the MC-PCA, open symbols are not. Color axis for oceans shows modern mean annual sea surface temperature 73,74. Land shown in gray. Map was created using ArcMap 10.7.1. ArcGIS and ArcMap are copyright Esri, www.esri.com.**

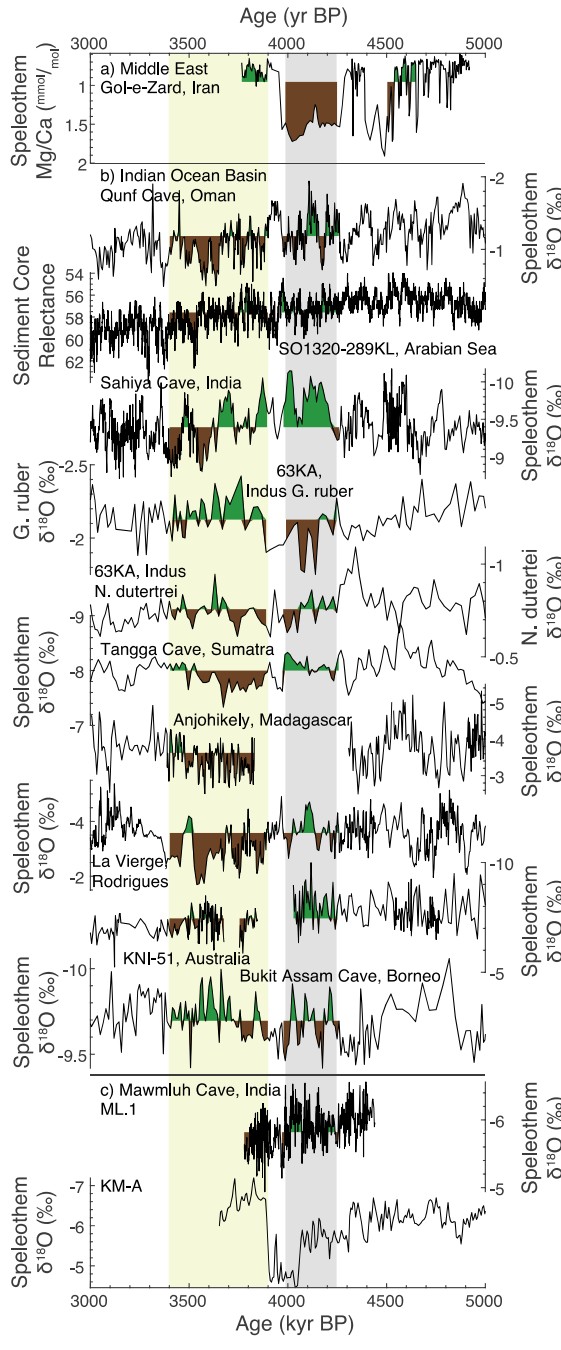

**Figure 2: Comparison of climatic shifts in the original records. a) Speleothem record of the 4.2 kyr event from the center of action in the Middle East. b) Indian Ocean basin hydroclimate records from 5000-3000 yr BP used in Monte-Carlo principal component analysis. c) speleothem records from Mawmluh Cave, India. Gray box indicates the timing of the 4.2 kyr event (4.26 to 3.97 kyr BP) as defined at Gol-E-Zard. Yellow box indicates the dry shift from PC1: 3.9-3.4 kyr BP. Green and brown shading indicate values corresponding to wetter and drier conditions than mean values for the records between 5000 and 3000 yrs BP within the shaded boxes.**

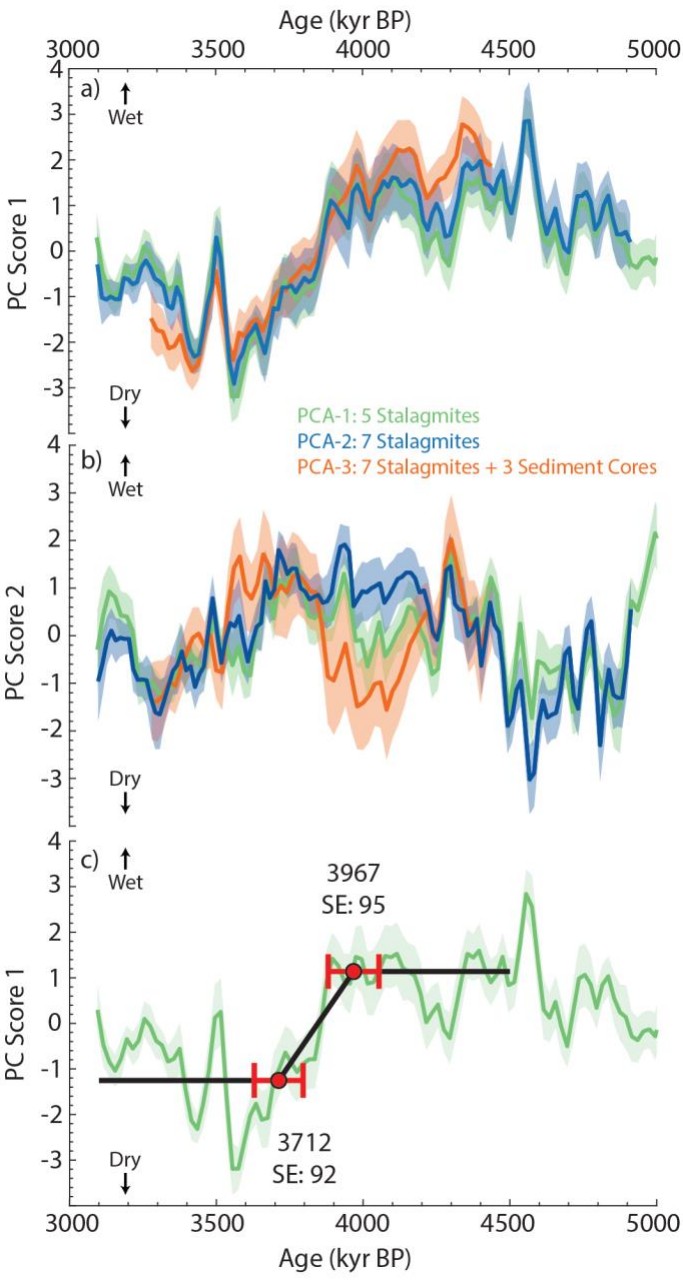

**Figure 3: Monte-Carlo Principal component analysis of Indian Ocean hydroclimate records: a) Principal Component 1, b) Principal Component 2. Color lines indicate mean principal component from 2000 individual analyses on redrawn age models, with 1 standard**
**deviation shading. c) RAMPFIT results of PCA-1, PC1 (black line with red error bars on ramp start and end points). Green: Five speleothem PCA, Blue: 7 speleothem PCA, Orange: 7 speleothem and 3 sediment core PCA.**





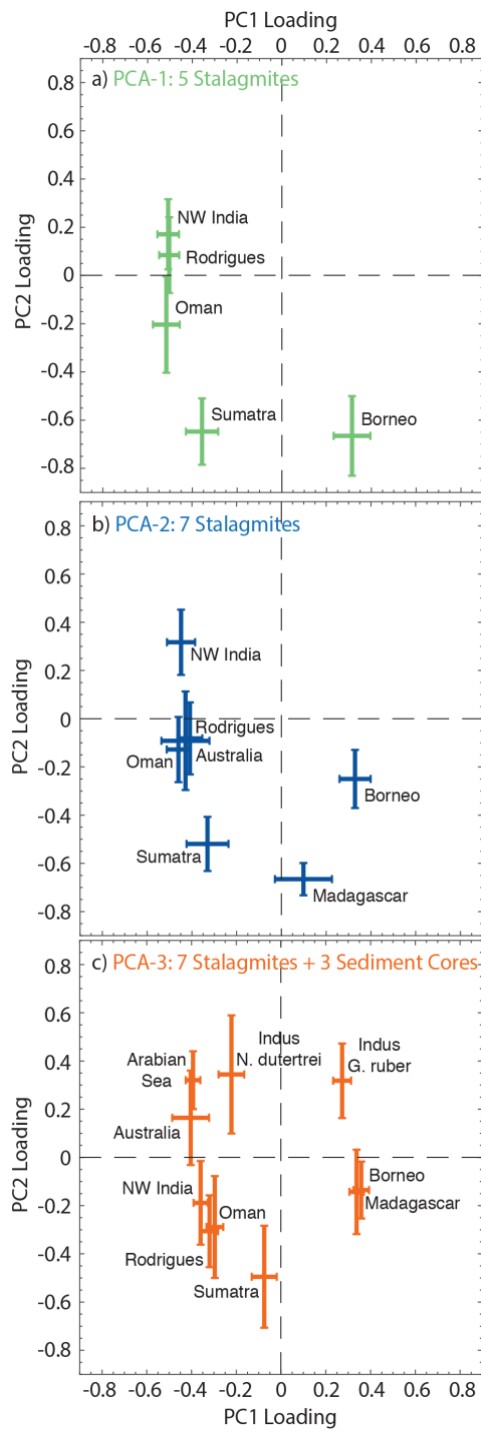

**Figure 4: Loadings of individual hydroclimate records on the first two principal components for each MC-PCA analysis with 1 standard deviation error bars. a) PCA-1, b) PCA-2, c) PCA-3.**



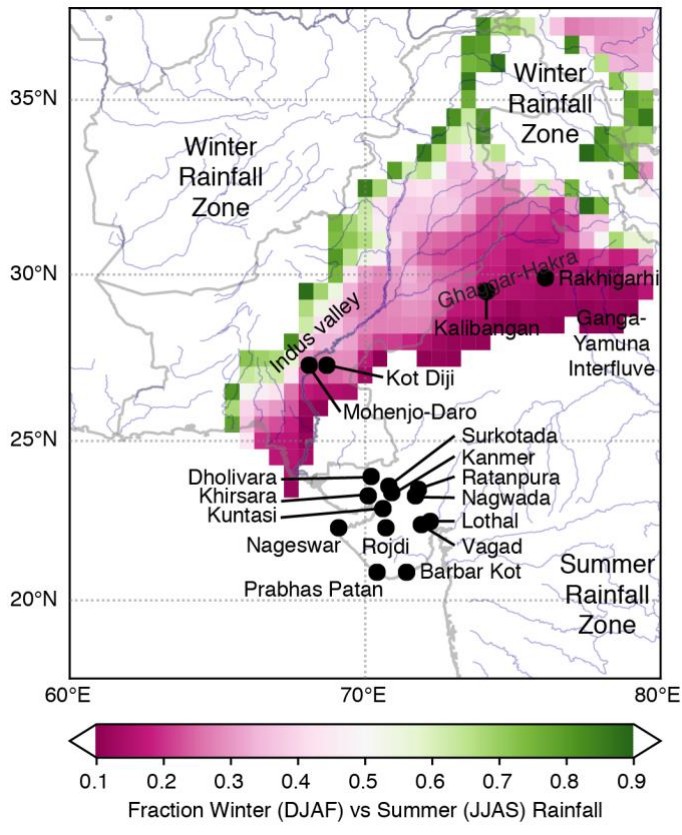


**Figure 5: Location map of significant archaeological sites (black dots) with multiple radiocarbon dates. Color axis (pink to white to green) shows the fraction of rainfall falling during the winter (DJAF) season compared to the summer (JJAS season) between 1981 and 2010 using Global Precipitation Climatology Centre (GPCC) Full Data Reanalysis Version 7 0.5°x0.5° Monthly Totals (Schneider et al., 2015). Areas with greater than 90% rainfall falling in summer compared to winter, or vice-versa are left white and**
**labelled.**

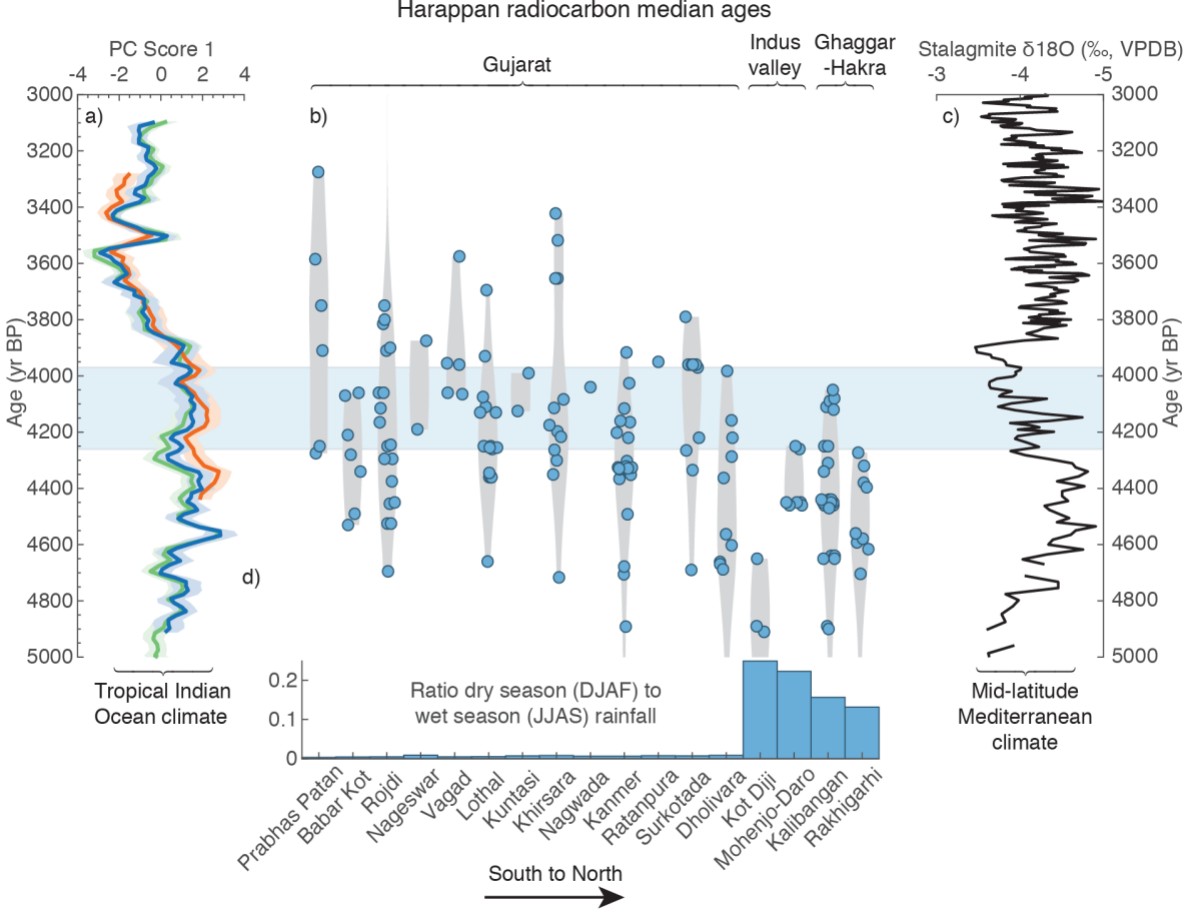

**Figure 6: Climatic changes and the Harappan civilization between 5 and 3 kyr BP. a) results from the Monte-Carlo principal component analysis of Indian Ocean hydroclimate records. PC1 from the 5 speleothem PCA (green), 7 speleothem PCA (blue) and 7 speleothems plus 3 sediment core PCA (orange) with 1 standard deviation shading. b) Violin plot of median ages of radiocarbon dates at major Harappan archaeological sites (blue dots) with kernel density estimate (grey shading). Plot adapted from Sengupta et al. (2020) with radiocarbon data from: Kalibangan, Kot Diji, Mohenjo-Daro (Brunswig, 1975), Prabhas Patan, Babar Kot, Nagwada, Surkotada, Raranpura, Kuntasi, Nagwada, Vagad, Lothal, Nageswar (Herman, 1996), Rojdi (Brunswig, 1975; Herman, 1996), Kanmer (Pokharia et al., 2011), Rakhigarhi (Nath et al., 2014; Vahia et al., 2016), Khirsara (Pokharia et al., 2017), Dholivara (Sengupta et al., 2020). c) Stalagmite RL4 from Renella cave, Italy ((Zanchetta et al., 2016), (Isola et al., 2019), using a 2018 updated chronology available in the SISAL database (Atsawawaranunt et al., 2018). d) Ratio of mean rainfall (1981-2010, GPCC V7 global gridded dataset at 0.5° × 0.5° resolution (Schneider et al., 2015)) falling in winter (DJAF) vs summer (JJAS) at each archaeological site. Blue horizontal box indicates the timing of the 4.2 kyr event as determined in Carolin et al., 2019 (4.26–3.97 kyr BP).**





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
