# Peer review of "Circum-Indian ocean hydroclimate at the mid to late Holocene transition: The Double Drought hypothesis and consequences for the Harappan"

_Climate of the Past, 2020_

## Short Comment (SC1) · 27 Nov 2020

Hi, The Harappan civilization was not the only civilization to experience climate change and subsequent desertification. All of North Africa and the Peninsula were savannahs in the Pharaohs' time. Today the accounts of the ancient Greeks and Romans recall this ecological disaster of great magnitude. At this time it was 2 $^\circ$ degrees higher, which increased the general hydrometry. The monsoon was rising higher, towards Pakistan and the Arabian Peninsula. Below are some stories that describe this climate change: https://nantt44.wordpress.com/2019/02/19/chapter-viiibetius-charmutha-arabia-felix-and-the-climatic-change/ Cordially.
* * *
[Figure]

**Fig. 1.**

[Figure]

---

## Editor Comment (EC1) · Denis-Didier Rousseau (Editor) · 27 Nov 2020

Reply to B Lepoivre

Thanks you for participating to the discussion phase of this manuscript. However I must remind you the aim of the discussion phase: "The discussion phase represents a unique opportunity to engage in an iterative and developmental reflective process. During this phase interactive comments can be posted by designated referees (anonymous or named) and all interested members of the scientific community (named), an

option that you chose. All participants are encouraged to stimulate further deliberation rather than simply to defend their position. This enhancement lead process is offered to maximize the impact of the article." All the very best

Denis-Didier Rousseau

CP co-editor in chief

---

## Short Comment (SC2) · 1 Dec 2020

The manuscript by Scroxton et al. presents a principal component analysis of regional records to understand the spatial pattern of precipitation changes over the 4.2 ka BP event, and the mid-late Holocene transition more broadly. While this analysis is useful and valuable, we wanted to draw attention to some specific errors in this manuscript up for review, so that they might get fixed for the final version. For full disclosure, we are the authors of the Giesche et al., 2019 paper from marine core 63KA (published in this journal, Climate of the Past), which is one of the ten records used in the PCA analysis

conducted by Scroxton et al.

Our points can best be illustrated by lines 351-357 of the Scroxton et al. manuscript, where the authors write: "The Double Drought hypothesis is consistent with the timing, location, and long-lasting nature of societal change during the decline and abandonment of the Harappan civilization. While this hypothesis is consistent with many paleoclimate records from the Indian subcontinent, it is not always consistent with their prior interpretation. For example, our hypothesis is in direct contrast to that of Giesche et al. (2019) who used foraminifera $\delta$18O records (interpreted as Arabian Sea salinity changes) to hypothesize a temporary reduction in summer rainfall at 4.2 kyr BP, and a longer lasting reduction in winter rainfall at 3.9 kyr BP. To reconcile these ideas, paleohydroclimate proxies with less ambiguous seasonality than speleothems and foraminifera will be required." (Scroxton et al., lines 351-357)

In contrast to what Scroxton et al. suggest here, Giesche et al., 2019 did not interpret the 63KA record only in terms of salinity changes. Using G. ruber foraminifera to reconstruct sea surface salinity from Indus River discharge was one component of the research (and linking this to a decline in Indian Summer Monsoon at 4.2 ka BP was originally proposed by Staubwasser et al., 2003). However, the main new research contributed by Giesche et al., 2019 was a reconstruction of upper ocean mixing (temperature gradients between species of foraminifera living at different depths), which relates to winter-driven surface evaporation and strength of the Indian Winter Monsoon. The 63KA core is thereby well-suited to resolve seasonal information using two different proxies in the same core. In summary, the Giesche et al., 2019 paper supports a decline in summer rainfall based on a decrease in Indus River freshwater discharge peaking at 4.1 ka BP, and suggests that a pronounced shift from plentiful winter rainfall at 4.3 ka BP to winter drought by 4.1 ka BP led to what we call the "4.2 ka event". Our main difference to the Scroxton et al. interpretation is that we suggest that summer rainfall largely recovered after 3.9 ka BP, whereas winter rainfall does not appear to make a recovery for at least several more centuries.

More specifically, Giesche et al., did not hypothesize that a winter drought began after 3.9 kyr BP as suggested by Scroxton et al. Rather, Giesche et al., 2019 write clearly about a winter and summer drought over the Indus Region by 4.1 ka BP, and specifically link the decrease in winter rain to impacts on the Indus Civilization. Therefore, it is important that the paper at hand is amended to note that Giesche et al. 2019 previously suggested a version of the double drought hypothesis for Indus Civilization (Harappan) decline in the Indus region.

There are several relevant sections to point to in the abstract and discussion of the Giesche et al., 2019 publication, which are succinctly summarized by a quote from the conclusion of that paper: Conclusion: "We propose that a combined weakening of the IWM and ISM at 4.1 ka led to what has been termed the "4.2 ka BP" drought over northwest South Asia. The intersection of both a gradually weakening ISM since 4.8 ka and a maximum decrease in IWM strength at 4.1 ka resulted in a spatially layered and heterogeneous drought over a seasonal to annual timescale. Regions in the western part of the Indus River basin accustomed to relying mainly on winter rainfall (also via river runoff) would have been most severely affected by such changes." (Giesche et al., 2019)

We hope the authors of the manuscript will consider our comments, amend the existing errors, and acknowledge the previous research on the topic of a double drought hypothesis with implications for Indus Civilization decline.

Nevertheless, the specific timing and during of seasonal droughts in our conclusions continue to differ: Scroxton et al. attribute a first drying episode at 4.26 ka BP to winter drought and note a second prolonged summer drought after 3.97 ka BP, while Giesche et al. (2019) found a gradual drying trend in summer monsoon after 4.8 ka BP that peaked by 4.1 ka BP along with a pronounced flip from abundant winter rain to a winter drought from 4.3 to 4.1 ka BP. Despite these differences, we find it broadly encouraging that Scroxton et al.'s PCA analysis of regional paleoclimate records supports the idea that droughts in both seasons of precipitation played an important role in this region

over the mid-late Holocene transition. This makes the case even more compelling for continued research into this topic from highly resolved and well-dated records with multiple proxies that can differentiate between seasonality.

References:

Giesche, A., Staubwasser, M., Petrie, C., & Hodell, D. (2019). Indian winter and summer monsoon strength over the 4.2 ka BP event in foraminifer isotope records from the Indus River delta in the Arabian Sea. Climate of the Past, 15 (1), 73-90. https://doi.org/10.5194/cp-15-73-2019

Staubwasser, M., Sirocko, F., Grootes, P. M., & Segl, M. (2003). Climate change at the 4.2 ka BP termination of the Indus valley civilization and Holocene south Asian monsoon variability. Geophysical Research Letters, 30(8). https://doi-org.ezp.lib.cam.ac.uk/10.1029/2002GL016822

---

## Editor Comment (EC2) · Denis-Didier Rousseau (Editor) · 2 Dec 2020

Dear Alena,

Thank you very much for taking the opportunity offered to the community by the discussion phase to comment this paper submitted by Nick Scroxton. Your comment is clearly scientifically sound and should be very useful for improving the first version of the manuscript. I am therefore strongly encouraging the authors to reply you as soon as possible.

All the very best

denis-didier Rousseau

CP co-editor in chief

---

## Author Comment (AC1) · 10 Dec 2020

This comment is correct, we made a mistake in summarising the differences between our new synthesis and the results of Giesche et al, (2019) and have ended up mis-representing their paper. Giesche et al., 2019 does indeed propose a reduction in the winter monsoon around 4.2 kyr BP. They also propose a similar 'double drought' idea, suggesting that this winter drought occurred alongside a summer decrease in rainfall, although there are considerable differences in what climate signals are determined droughts. We are happy to correct our work and take the opportunity to discuss in

more detail and more nuance, where our interpretations agree, and where they do not.

We feel that largely the two studies agree, as might be expected as Giesche et al (2019) provide two of the records used in our composite. A secular trend of decreasing summer rainfall, an increase in winter rainfall at 4.6-4.5 kyr BP, a winter rainfall drought around the 4.2 kyr event, an increase in rainfall between 3.5 and 3.0 kyr BP are all consistent. Where the two studies disagree are in the timing of the summer rainfall drought. Giesche et al's double drought is a combination of winter rainfall drought on top of a secular trend of decreasing summer rainfall, perhaps with an slight acceleration and peak at around 4.2 kyr BP – ie. two simultaneous droughts. While we do not doubt the secular trend in decreasing summer rainfall, our analysis suggests that an acceleration in the summer rainfall decline began later at 3.97 kyr BP, peaking at 3.6 kyr BP. Our double drought is therefore sequential rather than simultaneous: winter then summer droughts, both superimposed on the long term drying trend.

We propose making the following changes to the manuscript: 1) Citing Giesche et al 2019 on the winter monsoon paragraph of section 4.3 (Line 271) 2) Citing Giesche et al 2019 on the winter rainfall drought leading to harsher stress on more northerly Harappan sites (Line 299) 3) Replacing the two paragraphs that end the section 4.4 discussion (lines 352-362) with the following:

"The Double Drought hypothesis builds on recent high resolution paleoclimate work in the region. Both Giosan et al. (2018) and Giesche et al. (2019) proposed an increase in winter rainfall at 4.5 kyr BP and suggested that the increase plays a role in the flourishing of the Mature Urban phase of the Harappan civilisation which starts coincidentally. While we don't interpret either PC1 or PC2 as direct measures of winter rainfall, both show an increase (wetting signal) between 4.6 and 4.5kyr BP, corroborating their findings. The traditional view of Indian subcontinental rainfall at the mid-late Holocene transition is that the 4.2 kyr event represents a decline in summer monsoonal rainfall. However, Giesche et al. (2019) indicated an additional winter rainfall drought starting around 4.3 kyr BP and peaking at 4.1 kyr BP. Giesche et al. (2019) also inferred that a

winter drought would have impacted the more northerly Indus valley civilisation centres more heavily than those in Gujarat. Giosan et al., (2019), however, proposed that the period of increased winter rainfall lasted to 3.0 kyr BP. PC1 in our analysis is interpreted as a summer monsoon proxy and so does not directly measure or infer winter rainfall. However, the absence of a signal in the likely summer rainfall dominated PC1 agrees with the idea of a winter rainfall drought dominating the region at this time. Similarly, the PCA does not allow for firm conclusions about whether winter rainfall returns at 3.9 kyr BP. The most likely mechanism for winter drought, a change in westerly disturbances from the Mediterranean forced by the 4.2 kyr event came to an abrupt end at 3.9 kyr BP, and therefore a return of abundant winter rainfall is likely, presumably continuing to 3.0 kyr BP (Giosan et al., 2019).

Giesche et al, (2019) also proposed a decrease in summer rainfall at the time, as part of a long-term trend since at least 4.8 kyr BP, peaking at 4.2kyr BP, with a return to pre-excursion values by 3.0 kyr BP. There is little doubt that there was a gradual decrease in monsoonal rainfall, likely resulting from decreasing northern hemisphere insolation over the course of the Holocene. Our PC1, and several individual records also agree with at least a partial recovery by 3.0 kyr BP. Our synthesis disagrees with Giesche et al., (2019) on the timing of the summer rainfall drying on top of the secular trend. Giesche et al (2019) suggest shortly after 4.2kyr BP, i.e. a simultaneous double drought. Whereas the regional PCA suggests drying beginning at 3.97 and peaking at 3.6kyr BP, i.e. a sequential double drought. As discussed in Giesche et al., (2019) and Section 4.2 above, numerous hydroclimate paleoclimate records from the region support one or more of the three droughts (secular trend, 4.2-3.9kyr BP anomaly and 3.9kyr BP step change). To reconcile the exact timing of these rainfall fluctuations and their seasonality, further study is required with paleohydroclimate proxies with less ambiguous seasonality than speleothems and foraminifera.

Nether the less, our Double Drought Hypothesis is consistent with broader climatic changes in 'upstream' areas of Indus valley rainfall; changes in winter mid-latitude

westerlies are interpreted as influencing winter rainfall, while changes in tropical Indian Ocean hydroclimate are interpreted as influencing summer monsoonal rainfall."

---

## Referee Comment (RC1) · Anonymous Referee #1 · 29 Dec 2020

Scroxton et al. present an age-uncertain MC-PCA analysis of 10 speleothem records and 3 sediment records from the circum Indian Ocean basin, and discuss the results in the context of hypotheses driving the decline of the Mature Harappan period of the Harappan civilization. A specific focus of the discussion is evidence for a "4.2 kyr event", which has been hypothesized to drive the decline of the Harappan civilization.

The paper is well-written, presents an interesting new analysis, and has the potential to be an important contribution to the literature. I do have several major points that need to be addressed before I can recommend the manuscript for publication:

[Figure]

Major points

1. The MC-PCA analysis is state-of-the-art, but the spatial component of the results go largely unpresented or discussed. This is especially the case for PC2, given the mix of loadings, but even the spatial patterns of the strength of PC1 could shed light on the question at hand. I suggest the authors add maps that illustrate the PC1 and 2 loadings, and their uncertainties, and discuss their results. This is particularly important for PC2. The authors describe this as a dipole that represents a fluctuating wet and dry signal. The spatial structure of this dipole, especially as it relates the the Harappan civilization, needs to be presented and interpreted; including a discussion of the age-uncertainty, and record selection uncertainties (PC-1,-2,-3).

2. From the analysis, primarily the gradual decline of the summer monsoon recorded by PC1, the authors propose a "Double-drought" hypothesis, that suggest that the combination of a shorter term (300-yr-long) winter precipitation drought, followed by the monsoon weakening beginning around 3.9 kyr weakening summer precipitation may have driven the collapse. This hypothesis is reasonable, and is supported by the MC-PCA analyses, a synthesis of Harappan archeological sites, and a single speleothem record from Italy (RL4) interpreted to illustrate the decline in winter precipitation in India. The contrast between the first two lines of evidence, and the Mediterranean speleothem is striking, especially given the importance of winter precipitation in the double-drought hypothesis. Unlike the analysis that went into summer precipitation, only a single record (RL4) is shown to draw inference about "Mid-latitude Mediterranean Climate", with a notably different age model than that used in the original (2016) publication. To defend the "Double-Drought" hypothesis, the authors need to better characterize Mediterranean climate, and its relation to winter precipitation on the Indian subcontinent. The paragraph from 359-361 states that this is broadly consistent, but this needs to be robustly established. I recognize that this is a significant request, but it would be ideal to see an analysis, either new or from the literature, of multiple records from Mediterranean (if that is indeed the best way to estimate winter precipitation in India),

that handles age uncertainty and disagreement between records. As is, it's difficult to evaluate the hypothesis when this key line of reasoning is so poorly supported.

3. In new studies that rely on syntheses of multiple records, it's critical that the data (and ideally the code) used to conduct the analyses is available, and replicable. Data that are only available upon request are not publicly accessible, and I strongly encourage the authors to archive the data used in the analysis in a public data repository. If they cannot, the authors need to explain why they cannot follow the best practice recommendation in the "Data availability" section.

4. Why is PC-3 truncated? Looking at the data in figure 2, the three sedimentary records span the full length of PC-1, and PC-2. This decision needs to be explained and justified.

Detailed points:

38-39: Awkward to say "increasingly recognized", and then have only references from 2003 and 2004.

115: "Between the records"? Do you mean between the ages?

116: Upscaled. Might be better to say "degraded".

140-144: Why? It seems easy enough to use identical ensemble members each time through. More importantly, can you justify treating multiple records from the same core as independent climate records?

172: I suggest you start by describing this as a dipole, and then describe it as fluctuating wet and dry.

178: I don't think you can call this a dry period in PC2, since it's a dipole, it must be dry some places and wet others. In general, the spatial characteristics of PC2 should be fleshed out.

294: 4.26 to 3.97 kyr BP this is a lot of precision, where do these numbers come from,

and what are their uncertainties? This is comment relevant throughout the discussion.

309: What does this analysis say about whether the mature Harappan occurred during a short term pluvial?

314: "was likely caused by a reduction in western disturbances that bring rainfall from the Mediterranean and Middle East (Figure 6c). " How do you know this?

Figure 1. Add Harappan sites Figure 5. Is this the ratio of winter to summer? Or the fraction that winter represents out of total rainfall. Also, an overview map showing where this is would be helpful for those less familiar with the region

Figure 7. The d) label is a little hard to locate. Consider moving it to the top left corner.

---

## Referee Comment (RC2) · Harvey Weiss (Referee) · 4 Jan 2021

Global KM-A Congruence and the Indus Collapse

Harvey Weiss

Department of Near Eastern Languages and Civilizations, Environmental Studies Program, and School of Environment, Yale University

The IUGS-recognized global boundary stratotype for the 4.2 - 3.9 ka BP event, marking

the middle to late Holocene transition to the Meghalayan stage, is the KM-A speleothem $\delta$18O record from Mawmluh Cave, Meghalaya, NW India, that is an Indian Summer Monsoon (ISM) drought record (Berkelhammer et al 2012; Walker et al 2019). The recent analysis of the Indus delta foraminifera record at core 63KA has identified, as well, the Indian Winter Monsoon drought synchronous with the 4.2 ka BP ISM drought (Giesche et al 2019). The global boundary sub-stratotype is the Mt. Logan Yukon glacial core's $\delta$18O moisture event (Fisher et al 2008).

Scroxton et al present a principal components analysis of seven recent $\delta$18O speleothem records from the Indian Ocean region and the Giesche et al 2019 delta foraminifera analyses (line 110) "to investigate the impacts of the 4.2 kyr event on tropical Indian Ocean basin monsoonal rainfall" and the late third millennium BC Indus urban collapses. Similar to earlier analyses using lake sediment records (e.g., Leipe et al 2014), Scroxton et al note the succession of two gradual centuries long dry periods separated by the 4.2 ka BP aridification event, but from their analysis present four new conclusions.

Scroxton et al conclude that the Mawmluh Cave KM-A speleothem is not a useful stratotype because (a) line 370 "The KM-A record replicates neither the other speleothem from Mawmluh Cave (Kathayat et al., 2018) nor any regional records (this study)" and (b) line 374 "is low resolution, low dating frequency, not replicable within its own locality, ambiguously defining a climate event that is not significant across its climate domain". Figure 1 presents the onset, terminus and resolution of the Mawmluh KM-A 4.2 ka BP event and some related proxies. As previously noted, Mawmluh KM-A is similar within standard deviations to the other Mawmluh speleothems ML.1, 2 in both onset and terminus (Kathayat et al 2018). It is similar, as well, to the records at the Indus delta (Giesche et al 2019), and to the sampling resolutions of the two recent ISM Madagascar speleothems (Wang et al 2019; Scroxton et al 2020 in review). Not listed here is the Sahiyah Cave, NW India speleothem that certainly does not present an abrupt 4.2 ka BP event, but was "manifest as an interval of declining ISM strength, marked by rel-

atively higher amplitude of del 18O variability and slow speleothem growth"(Kathayat et al 2017). Mawmluh KM-A is also congruent and synchronous with the high resolution speleothem westerlies proxy for the 4.2 ka BP dust/drought event at Gol-e Zard NW Iran (Carolin et al 2019) as shown in Figure 2 top, that is synchronous with the settlement collapse in northern Mesopotamia (Weiss et al 2012) and many regional settlement abandonments across the Mediterranean.

Secondly, Scroxton et al argue that the 4.2 ka BP event was of (line 400) "limited impact on tropical monsoonal rainfall around the circum-Indian Ocean basin". However, Scroxton et al do not include the Mawmluh KM-A speleothem record in their principal components analysis (line 364, "too short to be included") nor the ISM Tibetan plateau speleothem record (Cai et al 2012), and the possibly anti-phase Southeast African records (Humphries et al 2020). Most problematically, however, Scroxton et al ignore the Horn of Africa, the Ethiopian highlands, as there are no speleothem paleoclimate records. Nevertheless, the Ethiopian highlands are a major component, multiply recorded, of ISM sourced Indian Ocean basin hydroclimate. The 1200 mm of highland Ethiopian ISM precipitation collect at Lake Tana, and become the Blue Nile and Atbara Rivers, which together provide 90% per cent of Nile peak flow as measured by air mass back trajectories and 97% of Nile annual sediment load (Williams 2019: 28; Woodward et al 2014; Costa 2014). The Nile River extends 4759 kms from Lake Tana to the delta (William 2019: 117), or 2000 kms longer than the Indus, and was the primary physical determinant of ancient Egyptian irrigation agriculture. The sediment core from Lake Tana documents an important, albeit 200 year resolution, low stand at 4.2 ka BP (Marshall et al 2011) synchronous with other East African records that include the Lake Mega-Chad large scale dust mobilization (Kröpelin et al 2008; Francus et al 2013). This ISM/Nile source reduction, known at the Nile delta within numerous and various sediment core proxies, is recorded at the Nile deep sea fan marine cores. For example, the recent analyses of MD04-2276 include the 4.2 ka BP SST (Jalali et al 2017) and Mn/Al flux (Mologni et al 2020) events. Although there is interpolation across three radiocarbon dates, Figure 2 middle displays this SST event synchronous

and congruent with the Mawmluh KM-A record.

The significance of the 4.2 ka BP Nile event resides in its synchronism with the Old Kingdom collapse and the beginning of the First Intermediate Period (Barta 2019), one feature of which was considerable settlement abandonment at the Nile delta and resettlement in Middle Egypt. While the Old Kingdom collapse was also synchronous with rain-fed settlement abandonment across Mesopotamia and Syria at 2200 BC (Ramsey et al 2010; Weiss et al 2012), Upper Egypt and its Kerma culture, close to the source of Nile flow, did not experience similar Nile flow reductions nor regional settlement abandonments (Woodward et al 2014).

Thirdly, Scroxton et al state, (line 47) that "the areal extent of the 4.2 kyr BP event beyond the data-rich heartland of Mediterranean Europe (Bini et al 2019) and Mesopotamia (sic) (Kaniewski et al 2018) is unclear." But the event records are abundantly available. The 4.2 ka BP event extended to Australia (Deniston et al 2013) and to southern and Northern China, an ISM and East Asian Summer Monsoon (EASM) event, e.g., Hulun Lake (Zhang et al 2020), Dongshiya Cave (Zhang et al 2018), Lake Balikun (An et al 2011). At Lake Wuya in North China (Tan et al 2020) the event is recently described erroneously as gradual when $\delta18O$ increased and decreased abruptly at 4200 and 3996 ka BP. The EASM is, of course, ISM sourced (Liu et al 2015; Yang et al 2014), and a North Atlantic wavetrain for 50% of modern ISM drought events has now been identified (Buhar et al 2020).

Along the western Pacific the event is recorded in Japan (Park et al 2019) and in the Kuroshio Current's "Pulleniatina minimum event" (Zhen et al 2016; Zhang et al 2019; Shuhuan et al 2021) where its northeastern trajectory likely generated the Mount Logan Yukon 4.2 ka BP event (Fisher et al 2008). The Mount Logan 4.2 ka BP sub-stratotype, with 2-3 year resolution, is both synchronous and congruent with the Mawmluh KM-A event (Figure 1, Figure 2 bottom).

Across mid-latitude North America the event is also well documented, from western

Idaho to western Massachusetts, "with median moisture levels reaching a minimum from 4.2 to 3.9 ka" (Shuman and Marsicek 2016: 42; Shuman et al 2019) alongside the North American monsoon phase change recorded at Leviathan Cave (Lachniet et al 2020). In the South American Monsoon region, along the western coast of South America, northern sediment cores suggest abrupt wet eastern Cordillera events synchronous with abrupt dry western Cordillera and Altiplano events. Lake Titicaca, for example, experienced an abrupt diatom shift at ca. 4300 BP followed by a drought event from ca. 4200 - 3900 BP with a lake level drop of ca. 70 meters (Weide et al 2017). At the southernmost Andes, "Marcel Arevalo" caves MA 1-3 record a uniformly wet period from ca. 4.5 to 3.5 ka BP interrupted by an abrupt ca. 23% drop in precipitation centered at ca. 4.2 ka BP (Schimpf et al 2011), possibly associated with several volcanic eruptions.

Synchronously, the continental monsoon along Brazil's east coast and the South Atlantic Convergence Zone that crosses Brazil, experienced abrupt and radical alteration. The Lapa Grande speleothem's sharp, decreased spike in $\delta18O$ extended from ca. 4.2 – 3.9 ka BP (Stríkis et al 2011). At Chapada do Apodi, Northeastern Brazil, high resolution speleothems, clastic sediments and bat guano analyses display abrupt high $\delta13C$ and $\delta18O$ and low 87Sr/86Sr values indicating "a massive episode of soil erosion…the beginning of the Meghalayan chronozone, characterized as the aridification of this region, decline in soil production, drying out of underground drainages" (Utida et al 2020).

Apart from the dense distribution of Mediterranean and Western Asia records, including the Gol-e Zard speleothem congruent with Mawmluh KM-A, Figure 2 top, the 4.2 ka BP event synchronous records extend to Alpine Europe (e.g., Spannagel Cave, Fohlmeister et al 2012) and more than fifty subpolar North Atlantic records (Weiss 2019). The latter are now complemented by the high resolution north Iceland marine core MD99-2275 SPG event dated 4290±40 ka BP (Jalali et al 2019; Figure 1) and the Irminger Current event (McCave and Andrews 2019), which both suggest a 4.2 ka BP AMOC

slowdown. This was due, possibly, to the freshwater dosing associated with glacial melt documented synchronously, for example at the Agassiz glacial core (Vinther et al 2009; Fisher et al 2012; Lecavalier et al 2017).

Fourthly, Scroxton et al conclude, "The absence of a significant, widespread 4.2 kyr event in tropical Indian Ocean hydroclimate has consequences for the timing and causes of the deurbanization of the Harappan civilization in the Indus valley." The 4.2 - 3.9 ka BP event, line 270 "a winter rainfall drought" in their account, was line 31 "propagated from the Mediterranean and the Middle East" and "led to Harappan site abandonment in the Indus Valley", while settlement in Gujarat continued until "a more gradual but longer lasting reduction in summer monsoon rainfall beginning at 3.97 kyr BP".

When were the Harappan cities of the Indus Valley abandoned and when did Harappan period city and village Gujarat settlement terminate? The five Harappan cities that were abandoned and collapsed ("deurbanized"), are Mohenjo Daro (150 has., Sind), receiving ca. 180 mm ISM precipitation per annum (Abbas et al 2018), Ganwariwala (40-80 has., Cholistan), on the now dry Hakra River, Harappa (100 has. Punjab) adjacent to the Ravi effluent of the Indus, with ca. 50% ISM precipitation annually, Rakhigarhi (150-350 has., Haryana) on a Gaggar-Hakra system paleo-channel, and the exceptional Dholavira (120 has., Gujarat) on Khadir island in the Great Rann of Kutch. These were succeeded by habitat-tracking smaller Late Harappan settlements eastward from the Indus and Ghaggar-Hakra systems.

The standard periodization of Harappan occupations, from the fuzzy chronologies of the 70s and 80s, remains Harappan ca. 2600-1900 BC, or a 700 year period within which urban growth and then abandonment occurred before onset of the Late Harappan period at ca. 1900 BC (Shaffer 1992). When, and at what rate, the Harappan urban abandonments occurred during this 700 year gross ceramic definition period is yet uncertain. There are no useful data for the abandonment period at Mohenjo-Daro, and Ganweriwala remains unexcavated. Harappan occupation at Harappa was divided into 3 ceramic assemblage periods, with the end mature urban Harappan period 3C dated ca. 2200-1900 BC. That date, however, is derived from thirty-three phase 3 dates (Kenoyer 1991) and two phase 3C dates (Meadow and Kenoyer 1995), all long-lived wood charcoal samples, most distributed similar to period 2 dates, with only four or five that might extend beyond 2200 BC using an early calibration. It even seems that period 3C could end ca. 2200 BC, problematically around the time suggested it began even though 3 major building levels, yet unexcavated, were assigned to it (Meadow and Kenoyer 1995).

For the occupation at Rakhigarhi, Scroxton et al Figure 6b has nine radiocarbon dates. These are two Harappan period dates from unknown sample material, one at 4560±90 and one at 4320±90, with calibration uncertain (Nath 2014), and seven dates from wood charcoal of unknown stratigraphic proveniences (Vahia et al 2016). Four Late Harappan dates from Rojdi are presented by Scroxton et al Figure 6b from Herman 1996 Table 8. All are long–lived wood charcoal samples, two have no provenience and the two provenienced dates have three-digit standard deviations calibrated 2320-1900 BC and 800-200 BC. Four Rojdi Late Harappan dates are presented from Herman 1996, two have no provenience and the two provenienced dates have 3-digit standard deviations calibrated 2320-1900 BC and 800-200 BC. Dholavira now provides 2 otolith sample radiocarbon dates, one each from strata that bound the phase 5 abandonment for which there are no radiocarbon dates (Sengupta et al 2020). In summary, Scroxton et al should explain why their Figure 6b kernel density estimation of median radiocarbon dates of long-lived wood charcoal samples, with three-digit standard deviations, some from unknown strata, provide a chronology of Indus settlement abandonments.

Rakhigarhi, the largest Harappan city, "increased in size during the Mature Harappan period but was depopulated and then abandoned by the start of the Late Harappan period" (Nath et al 2014). Rafique Mughal's vanguard and prodigious regional survey recorded the highest density of Harappan settlement along the Pakistani Hakra (Mughal 1997). The abandonment data from a 15 km radius settlement survey around

Rakhigarhi, using Dewar's algorithm, indicates possible 80% abandonment (hectares occupied) sometime during the 700 year span of the Harappan period and before late Harappan period onset (Petrie and Lyman 2020), i.e., some presently undeterminable span near the end of the Harappan period. Rakhigarhi receives between 300 and 400 mm precipitation, sufficient for winter cereal agriculture, and its ISM fed Gaggar-Hakra paleochannel could have provided for ponds and flood water farming. The ISM provides 80% of annual precipitation at Karsandi, 120 kms SW of Rakhigarhi (Dixit et al 2018), and was disrupted by the 4.2 ka BP megadrought recorded at nearby Kotla Dahar paleolake which receives 75% ISM precipitation (Dixit et al 2014). The paleolake's abrupt and severe 4% $\delta$18O increase at ca 4100$\pm$100 ka BP was synchronous with the ISM Mawmluh KM-A record.

At Dholavira in Gujarat, the region that receives 309 mm annual precipitation, the ISM flooding of two streams that bounded the city north and south collected in large artificial reservoirs (Agrawal et al 2018). In spite of low resolution archaeological dating, the synchronous reduced ISM strength at 4.2 - 3.9 ka BP could account for the city's abandonment (Sengupta et al 2020). Scroxton et al Figure 6B, however, indicate continued occupation at Dholavira beyond the western Iran onset of the 4.2 ka BP event (Carolin et al 2019) synchronous with the KM-A ISM event, by plotting medians for two radiocarbon dates from end of phase IV and one from phase VI, when only the Dholavira castle was occupied. Phase V at Dholavira, however, is the end Harappan period abandonment and collapse followed by an occupational hiatus of uncertain duration (Sengupta et al 2020). In summary, Scroxton et al conclude mistakenly that the Indus River and Rakhigarhi, Gaggar-Hakra region abandonments / collapses were caused by a regionally insignificant but abrupt 4.2 ka BP IWM drought event and that the Dholavira abandonment and collapse were a continued occupation through the 4.2 ka BP event.

This drought event was the ISM century-scale drought recorded at Mawmluh KM-A, ML. 1, 2, and at region-wide circum-Indian Ocean basin speleothem, marine and lake

cores that include those for the 4.2 ka BP event in the Ethiopian highlands and Nile River. This occurred synchronously with the westerlies' 4.2 ka BP abrupt event reduction of Mediterranean, West and South Asian precipitation, the EASM events in China, the north Atlantic glacial, lake and marine core events, and the North and South American 4.2 ka BP events. Meanwhile, the archaeological evidence for the synchronous collapse of Egyptian, Mediterranean, West Asian, and Indus settlement systems at 4.2 ka BP appears increasingly robust.

REFERENCES

Abbas, Farhat, Rehman, Iqra, Adrees, Muhammed, Ibrahim, Muhammad, Saleem, Farhan, Ali, Shafaqat, Rizwan, Muhammad, Salik, Muhammad Reza. Prevailing trends of climatic extremes across Indus-Delta of Sindh-Pakistan, Theor Appl Climatol 131, 1101–1117, 2018. DOI 10.1007/s00704-016-2028-y

Agrawal, Silky, Majumder, Mantu, Bisht, Ravindra Singh, Prashant, Amit, Archaeological studies at Dholavira using GPR, Current Science, 114, 879-887, 2018.

An, Cheng-Bang An, Lu, Yanbin, Zhao, Jiajiu, Tao, Shichen, Dong, Weimaio, Li, Hu, Jin, Ming, Wang, Zongli, A high-resolution record of Holocene environmental and climatic changes from Lake Balikun (Xinjiang, China): Implications for central Asia, Holocene, 22, 43–52, 2011. DOI: 10.1177/0959683611405244

Bárta, Miroslav. Analyzing Collapse, the rise and fall of the Old Kingdom. NY, AUC Press, 2019.

Berkelhammer, M., Sinha, A., Stott, L., Cheng, H., Pausata, F. S. R., and Yoshimura, K.: An Abrupt Shift in the Indian Monsoon 4000 Years Ago, ed., Giosan, L., Fuller, D. Q., Nicoll, K., Flad, R. K., and Clift, P. D., American Geophysical Union AGU, Washington, D. C., 75–88, https://doi.org/10.1029/2012GM001207, 2012.

Borah, P.J., Venugopal, V., Sukhatme, J., Muddebihal, P., Goswami, B.N., Indian monsoon derailed by North Atlantic wavetrain, Science, 370, 1335–1338, 2020.

10.1126/science.aay6043

Cai, Yanjun, Zhang, H., Cheng, Hai, An, Z., Edwards, R. Lawrence, Wang, X., The Holocene Indian monsoon variability over the southern Tibetan Plateau and its teleconnections, EPSL, 335–336,135–144, 2012.

Costa, Kassandra, Russell, James,, Konecky, B., Lamb, Henry, Isotopic reconstruction of the African Humid Period and Congo Air Boundary migration at Lake Tana, Ethiopia, Quaternary Science Reviews, 83, 58-67, 2014. http://dx.doi.org/10.1016/j.quascirev.2013.10.031

Denniston, Rhawn, Wyywoll, K.-H., Polyak, Victor, Brown, J.R., Asmerom, Yemane, Wanamaker, A.D. Jr., LaPointe, Z., Ellerbroek, R., Barthelmes, M., Cleary, D., Cugley, J., Woods, D., Humphreys, W.F., A Stalagmite record of Holocene Indonesian–Australian summer monsoon variability from the Australian tropics. Quaternary Science Reviews, 78, 155-168, 2013. http://dx.doi.org/10.1016/j.quascirev.2013.08.004

Dixit, Yama, Hodell, David A., Petrie, Cameron A., Abrupt weakening of the summer monsoon in northwest India ∼4100 yr ago, Geology, 42, 339–342, 2014. doi:10.1130/G35236.1

Dixit, Yama, Hodell, David A., Giesche, Alena, Tandon, Sampat K., Gázquez, F., Saini, Hari S., Skinner, Luke C., Mujtaba, Syed A.I., Pawar, Vikas, Singh, Ravinda, Petrie, Cameron A., Intensified summer monsoon and the urbanization of Indus Civilization in northwest India, Scientific Reports, 8, :4225, 2018. 10.1038/s41598-018-22504-5

Fisher, David, Osterberg, E., Dyke, A., Dahl-Jensen, D., Demuth, M., Zdanowicz, C., Bourgeois, J., Koerner, R.M., Mayewski, P., Wake, C., Kreutz, K., Steig, E., Zheng, J., Yalcin, K., Goto-Azuma, K., Luckman. B., Rupper, Summer, The Mt Logan Holocene–late Wisconsinan isotope record: tropical Pacific–Yukon connections, Holocene, 18, 667–677, 2008.

Fisher, David, Zheng, James, Burgess, David, Zdanowicz, Christian, Kinnard,

Christophe, Sharp, Martin, Bourgeois, Jocelyne, Recent melt rates of Canadian arctic ice caps are the highest in four millennia, Global and Planetary Change, 84-85, 3–7, 2012. http://dx.doi.org/10.1016/j.gloplacha.2011.06.005

Fohlmeister, Jens, Vollweiler, N., Spötl, Christoph, Mangini, A., COMNISPA II: Update of a mid-European isotope climate record, 11 ka to present, Holocene, 23, 749-754, 2012. https://doi.org/10.1177/0959683612465446

Francus, Pierre, von Suchodoletz, H., Dietze, M., v. Donner, R., Bouchard, F., Roy, A.-J., Fagot, M., Verschuren, D., Kröpelin, Stefan, Varved sediments of Lake Yoa (Ounianga Kebir, Chad) reveal progressive drying of the Sahara during the last 6100 years, Sedimentology, 60, 911–934, 2013.

Giesche, A., Staubwasser, M., Petrie, C., Hodell, D., Indian winter and summer monsoon strength over the 4.2 ka BP event in foraminifer isotope records from the Indus River delta in the Arabian Sea, Clim Past, 15, 73-90, 2019. https://doi.org/10.5194/cp-15-73-2019

Herman, C.G., "Harappan" Gujarat: The Archaeology-Chronology Connection, Paléorient, 22, 77-112, 1996.

Humphries, Marc, Green, Andrew, Higgs, C., Strachan, K., Hahn, A., Pillay, L., Zabel, N., High-resolution geochemical records of extreme drought in southeastern Africa during the past 7000 years, Quaternary Science Reviews, 236, 106294, 2020. https://doi.org/10.1016/j.quascirev.2020.106294

Jalali, Bassem, Sicre, Marie-Alexandrine, Kallel, Nejib, Azuara, J., Combourieu-Nebout, Nathalie, Bassetti, M.-A., Klein, V., High-resolution Holocene climate and hydrological variability from two major Mediterranean deltas (Nile and Rhone), Holocene, 1–11, 2017. DOI: 10.1177/0959683616683258

Jalali, Bassem, Sicre, Marie-Alexandrine, Azuara, J., Pellichero, V., Combourieu-Nebout, Nathalie, Influence of the North Atlantic subpolar gyre circulation on the 4.2

kaBP event, Clim. Past, 15, 701–711, 2019. https://doi.org/10.5194/cp-15-701-2019.

Jiang, Hui, Muscheler,R., Björck, S., Seidenkrantz,M.-S., Olsen, J., Sha, Longbin, Sjolte, J., Eiríksson, J., Ran, Liohua, Knudsen, Karen-Luise, Knudsen, Mads F., Solar forcing of Holocene summer sea-surface temperatures in the northern North Atlantic, Geology, 43, 203–206, 2015. https://doi.org/10.1130/G36377.1

Kathayat, Gayatri, Cheng, Hai, Sinha, Ashish, Yi, Liang, Li, Xianglei, Zhang, Haiwei, Li, Hangying, Ning, Youfeng, Edwards, R. Lawrence, The Indian monsoon variability and civilization changes in the Indian subcontinent, Sci. Adv., 2017;3: e1701296, 2017.

Kathayat, Gayatri, Cheng, Hai, Sinha, Ashish, Berkelhammer, Max, Zhang, Haiwei, Duan, Pengzhen, Li, Hanying, Li, Xianglei, Ning, Youfeng, Edwards, R. Lawrence, Evaluating the timing and structure of the 4.2 ka event in the Indian summer monsoon domain from an annually resolved speleothem record from Northeast India, Clim. Past, 14, 1869–1879, 2018 https://doi.org/10.5194/cp-14-1869-2018

Kenoyer, Jonathan Mark, Urban Process in the Indus Tradition: A Preliminary Model from Harappa. Harappa Excavations 1986-1990, A Multidisciplinary Approach to Third Millennium Urbanism, ed. Richard H. Meadow, Monographs in World Archaeology No.3, 29-60, Madison WI, Prehistory Press, 1991.

Kröpelin, Stefan, Verschuren, D., Lézine, A.-M., Eggermont, J., Cocquyt, C., Francus, P., Cazet, J.-P., Fagot, M., Rumes, B., Russell, J.M., Darius, F., Conley, D.J., Schuster, M., von Suchodoletz, H., Engstrom, D.R., Climate-Driven Ecosystem Succession in the Sahara: The Past 6000 Years, Science, 320, 765-768. 2008. 10.1126/science.1154913

Lachniet, M. S., Asmerom, Y., Polyak, V., Denniston, R., Great Basin paleoclimate and aridity linked to Arctic warming and tropical Pacific Sea surface temperatures, Paleoceanography and Paleoclimatology, 35, e2019PA003785, 2020. https://doi.org/10.1029/2019PA003785

[Figure]

Lecavalier, Benoit S., Fisher, David A., Milne, Glenn A., Vinther, Bo M., Tarasov, Lev, Huybrechts, Philippe, Lacelle, Denise, Main, Brittany, Zheng, James, Bourgeois, Joce-lyne, Dykeh, Arhtur S., High Arctic Holocene temperature record from the Agassiz ice cap and Greenland ice sheet evolution, Proc Natl Acad Sci, 114, 5952-5957, 2017.

Leipe, Christian, Demske, Dietere, Tarasov, Pavel E., HIMPAC Project Members, A Holocene pollen record from the northwestern Himalayan Lake Tso Mriri: Implications for palaeoclimatic and archaeological research, Quaternary International, 348, 93-112, 2014. http://dx.doi.org/10.1016/j.quaint.2013.05.005

Liu, Jinbao, Chen, Jianhui, Zhang, Xiaojian,Li, Yu, Rao, Zhiguo, Chen, Fahu, Holocene East Asian summer monsoon records in northern China and their inconsistency with Chinese stalagmite $\delta$18O records, Earth Science Reviews, 148, 194-208, 2015. https://doi.org/10.1016/j.earscirev.2015.06.004 Liu, Z., Yoshimura, K., Bowen, G.J., Buenning, N.H., Risi, C., Welker, J.M., Yuan, F., Paired oxygen isotope records reveal modern North American atmospheric dynamics during the Holocene, Nature Commu-nications, 5, 3701, 2014. doi:10.1038/ncomms4701

Marshall, Michael H., Lamb, Henry F., Huws, Dei, Davies, Sarah J., Bates, R., Bloemendal, Jan, Late Pleistocene and Holocene drought events at Lake Tana, the Blue Nile, Global and Planetary Change 78, 147–161, 2011. doi:10.1016/j.gloplacha.2011.06.004

McCave, I.N., Andrews, Distinguishing current effects in sediments delivered to the ocean by ice. II. Glacial to Holocene changes in high latitude North At-lantic upper ocean flows, Quaternary Science Reviews, 223, 105902, 2019. https://doi.org/10.1016/j.quascirev.2019.105902

Meadow, Richard and Jonathan Mark Kenoyer, Excavations at Harappa 2000-2001, new insights on chronology and city organization, South Asian Archaeology 2001, ed. J.-C. Jarrige and C. Lefèvre. Paris, CNRS. 207-225, 2005.

Mologni, Carlo, Revel, Marie, Blanchet, Cecile, Bosch, Delphine, Develle, Anne-Lise, Orange, François, Bastian, Luc, Khalidi, Lamya, Ducassou, Emmanuelle, Migeon, Sebestien, Frequency of exceptional Nile flood events as an indicator of Holocene hydro-climatic changes in the Ethiopian Highlands, Quaternary Science Reviews, 247, 106543, 2020. https://doi.org/10.1016/j.quascirev.2020.106543.

Mughal, M. Rafique. Ancient Cholistan: Archaeology and Architecture, Lahore, Feroz-sons, 1997.

Nath, A., Garge, T., Law, R., Defi̧ning the Economic Space of Harappan Rakhigarhi: an Interface of Local Subsistence Mechanism and Geologic Provenience Studies, Puratattva 44, 83–100, 2014.

Newby, P. E., Shuman, B.N., Donnelly, J.P., Karnauskas, K.B., Marsicek, J., Centennial-to-millennial hydrologic trends and variability along the North Atlantic Coast, USA, during the Holocene, Geophys. Res. Lett., 41, 4300–4307, 2014. doi:10.1002/2014GL060183.

Park, Jungjae, Park, Jinheum, Yi, Sangheon, Cheul Kim, Jin, Lee, Eunmi, Choi, Jieun, Abrupt Holocene climate shifts in coastal East Asia, including the 8.2 ka, 4.2 ka, and 2.8 ka BP events, and societal responses on the Korean peninsula, Sci Rep, 9, 10806. 2019. doi.org/10.1038/s41598-019-47264-8.2019

Petrie, Cameron A., Lynam, Frank, Revisiting Settlement Contemporaneity and Exploring Stability and Instability: Case Studies from the Indus Civilization, Journal Field Archaeology, 45, 1-15, 2020. 10.1080/00934690.2019.1664848

Ramsey, Christopher Bronk, Dee, Michael W., Rowland, Joanne M., Higham, Thomas F.G., Harris, Stephen A., Brock, Fiona, Quiles, Anita, Wild, Eva M., Marcus, Ezra S., Shortland, Andrew J., Radiocarbon-Based Chronology for Dynastic Egypt, Science, 328, 1554-1557. DOI: 10.1126/science.1189395

Schimpf, Daniel, Kilian, Rolf, Kronz, Andreas, Simon, Klaus, Spötl, Christoph, Wörner,

Gerhard, Deininger, Michael, Mangini, Augusto, The significance of chemical, isotopic, and detrital components in three coeval stalagmites from the superhumid southern-most Andes (53o S) as high-resolution palaeo-climate proxies. Quaternary Science Reviews, 30, 443-459, 2011.

Scroxton, Nick, Burns, Stephen J., McGee, David, Godfrey, Laurie R., Ranivo-harimanana, Lovasoa, Faina Peterson, Possible expression of the 4.2 kyr event in Madagascar and the southeast African monsoon, CP in review, 2020. https://doi.org/10.5194/cp-2020-137.

Sengupta, Torsa, Mukherjee, A., Bhushan, R., Ram, F., Bera, M.K., Raj, H., Dabhi, A., Bisht, R.S., Rawat, Y.S., Bhattacharya, S.K., Juyal, N. Sarkar, A., Did the Harappan settlement of Dholavira (India) collapse during the onset of Meghalayan stage drought? Journal of Quaternary Studies, 35, 382-395, 2020.

Shaffer, Jim G, The Indus Valley, Baluchistan and Helmand traditions: Neolithic through Bronze Age. Robert W. Ehrich, ed., Chronologies in Old World Archaeology. 3rd ed. Chicago, University of Chicago Press, 441–464, 425–446, 1992.

Shuhuan, Du, Xiang, Rong, Liu, Jianguo, Yan, Hongqiang, Sha, Longbin, Liu, J. Paul, Chen, Zhong, Ariful Islam, G.M., Herath, H.M. Dileep Bandara Herath, Variable Kuroshio Current intrusion into the northern South China Sea over the last 7.3 kyr, Palaeogeography, Palaeoclimatology, Palaeoecology, 562, 110093, 2021. https://doi.org/10.1016/j.palaeo.2020.110093

Shuman, B.N., Marsicek, J., The structure of Holocene climate change in mid-latitude North America, Quaternary Science Reviews, 141, 38-51, 2016.

Shuman, Bryan N., Marsiceck, Jeremiah, Oswald, Wyatt Oswald, Foster, David R. Foster, Predictable hydrological and ecological responses to Holocene North Atlantic variability,. Proc Natl Acad Sci, 116, 5985–5990.2019. https://www.pnas.org/cgi/doi/10.1073/pnas.1814307116

Stríkis, Nicolás M., Cruz, Francisco W., Cheng, Hai, Karmann, Ivo, Edwards, R. Lawrence, Vuille, Mathiase, Wang, Xianfeng, de Paula, Marcos S., Novello, Valdir F., Auler, Augusto S., Abrupt variations in South American monsoon rainfall during the Holocene based on a speleothem record from central-eastern Brazil, Geology, 39, 1075–1078, 2011. doi:10.1130/G32098.1.

Tan, Liangcheng, Li, Yanzhen, Wang, Xiqian, Cai, Yanjun, Lin, Fangyuan, Cheng, Hai, Sinha, Ashish, Edwards, R. Lawrence, Holocene Monsoon Change and Abrupt Events on the Western Chinese Loess Plateau as Revealed by Accurately Dated Stalagmites, Geophysical Research Letters, 46, e2020GL090273, 2020. https://doi.org/10.1029/2020GL0902732020.

Ujiié, Y., Ujiié, H., Taira, A., Nakamura, T., Oguri, K., Spatial and temporal variability of surface water in the Kuroshio source region, Pacific Ocean, over the past 21,000 years: evidence from planktonic foraminifera, Marine Micropaleontology, 49, 335-364, 2003.

Utida, Giselle, Cruz, Francisco W., Santos, Roberto V., Sawakuchi, Andre O., Wang, Hong, Pessenda, Luis C.R., Novello, Valdir F., Vuille, Mathias, Strauss, Andre M., Borella, Ana Claudiaa, Stríkis, Nicolas M., Guedes, Carlos C.F., De Andrade, Fabio Ramos Dias, Zhang, Haiwei, Cheng, Hai, Edwards, R. Lawrence, Climate changes in Northeastern Brazil from deglacial to Meghalayan periods and related environmental impacts, Quaternary Science Reviews, 250, 106655, 2020. https://doi.org/10.1016/j.quascirev.2020.106655

Vahia, Mayank N., Kumar, Pankaj, Bhogale, Abhijeet, Kothari, D.C., Chopra, Sundeep Shinde, Vasant, Jadhav, Nilesh, Shastri, Ranvir, Radiocarbon dating of charcoal samples from Rakhigarhi using AMS, Current Science, 111, 27–28, 2016.

Vinther, B., Buchardt, S.L., Clausen, H.B., Dahl-Jensen, D., Johnsen, S.J., Fisher, D.A.,Koerner, R.M., Raynaud, D., Lipenkov, V., Andersen, K.K., Blunier, T., Rasmussen, S.O., Steffensen, J.P. and Svensson, A.M., Significant Holocene thinning of

the Greenland ice sheet, Nature, 515, 385-388, 2009.

Walker, Mike, Head, Martin J., Lowe, John, Berkelhammer, Max, Björck, Svante, Cheng,, Hai, Cwynar, Les C., Fisher, David, Gkinis, Vasilieos, Long, Antony, Newnham, Rewi, Rasmussen, Sune Olander, Weiss, Harvey, Subdividing the Holocene Series/Epoch: formalization of stages/ages and subseries/subepochs, and designation of GSSPs and auxiliary stratotypes, Journal of Quaternary Science, 34 173–186, 2019. DOI: 10.1002/jqs.3097

Wang, Lixin, Brook, George A., Burney, David A., Riavo, Ny, Voarintsoa, G., Liang, Fuyuan, Cheng, Hai, Edwards, R. Lawrence, The African Humid Period, rapid climate change events, the timing of human colonization, and megafaunal extinctions in Madagascar during the Holocene: Evidence from a 2m Anjohibe Cave stalagmite, Quaternary Science Reviews, 210, 136-153. 2019. https://doi.org/10.1016/j.quascirev.2019.02.004

Weide, D. Marie, Fritz, Sherilyn C., Hastorf, Christine A., Bruno, Maria C., Baker, Paul A., Guedrone, Stephane, Salenbiend, Wout, A ∼6000 yr diatom record of mid- to late Holocene fluctuations in the level of Lago Wiñaymarca, Lake Titicaca (Peru/Bolivia), Quaternary Research, 88, 179–192, 2017. doi:10.1017/qua.2017.49

Weiss, Harvey, Manning, Sturt W., Ristvet, Lauren, Mori, Lucia, Besonen, Mark, McCarthy, Andrew, Quenet, Philippe, Smith, Alexia Smith, Bahrani, Zainab, Tell Leilan Akkadian Imperialization, Collapse and Short-Lived Reoccupation Defined by High-Resolution Radiocarbon Dating, Seven Generations since the Fall of Akkad, ed. H. Weiss, Studia Chaburiensia 3, Wiesbaden, Harrassowitz, 163-192. 2012.

Weiss, Harvey. The 4.2 ka BP event in the northern North Atlantic. Clim. Past Discuss., https://doi.org/10.5194/cp-2018-162-RC2, 2019.

Williams, Martin, The Nile Basin: Quaternary Geology, Geomorphology and Prehistoric Environments, Cambridge, Cambridge University Press, 2019.

Woodward, Jamie, Macklin, Mark, Fielding, Laura, Millar, Ian, Spencer, Neal, Welsby, Derek, Williams, Martin, Shifting sediment sources in the world's longest river: A strontium isotope record for the Holocene, Nile, Quaternary Science Reviews, 130, 124-140. 2014. https://doi.org/10.1016/j.quascirev.2015.10.040

Yang, Xunlin, Liu, Jianbao, Liang, Fuyuan, Yuan, Daoxian, Yang, Yan, Lu, Yanbin, Chen, Fahu, Holocene stalagmite $\delta$18O records in the East Asian monsoon region and their correlation with those in the Indian monsoon region. Holocene, 24, 1657-1664, 2014. https://doi.org/10.1177/0959683614551222

Zhang, Na, Yang, Yan, Cheng, Zhao, Jingyao, Yang, Xunlin, Liang, Sha, Nie, Xudong, Zhang, Yinhuan, Edwards, R. Lawrence Edwards, Timing and duration of the East Asian summer monsoon maximum during the Holocene based on stalagmite data from North China. Holocene, 28, 1631–1641, 2018. DOI: 10.1177/095968.

Zhang, Y., Zhou, X., He, Y., Jiang, Y., Liu, Y., Xie, Z., Sun, L., Liu, Z., Persistent intensification of the Kuroshio Current during late Holocene cool intervals, Earth and Planetary Science Letters, 506, 335-364, 2019.

Zhang, Shengrui, Xiao, Jule, Xu, Qinghai, Wen, Ruilin, Fan, Jiawei, Huang, Yun, Li, Manyue, Liang, Jian, Contrasting impacts of the 8.2 and 4.2 ka abrupt climatic events on the regional vegetation of the Hulun Lake region in northeastern China. Journal of Quaternary Science, 35, 831–840, 2020. 10.1002/jqs.3231 2020.

Zhjeng, Xufeng, Li, Anchun, Kao, Shuh Ji, Gong, Xun, Frank, Martin, Kuhn, Gerhard, Cai, Wenju, Yang, Hong, Wan, Shiming, Zhang, Honghai, Jiang, Fuqing, Hathorn, Edmund, Chen, Zhong, Hui, Bangqi, Synchronicity of Kuroshio Current and climate system variability since the Last Glacial Maximum, EPSL, 452, 247–257, 2016.

[Figure]

| 4.2 ka BP proxy site | Onset | Terminus | Resolution | Publication |
|---|---|---|---|---|
| KM-A Mawmluh India | 4303±26 | 4071±31 | 5-6 | Berkelhammer et al 2012 |
| KM.1, 2 Mawmluh India | 4255±16 | 3916±13 | 1 | Kathayat et al 2018 |
| 63KA ISM Indus delta | 4194±30 | 3933±30 | 18 | Giesche et al 2019 |
| 63KA IWM Indus delta | 4266±30 | 3954±30 | 18 | Giesche et al 2019 |
| AK-1 Madagascar | 4310±34 | 3830±54 | 5 | Scroxton et al (in review) |
| ANJ94-5 Madagascar | 4340±30 | 3990±10 | 10 | Wang et al 2019 |
| Gol-e Zard NW Iran | 4260±40 | 3970±70 | 2-15 | Carolin et al 2019 |
| MD99-2275 N Iceland | 4290±40 | 4060±40 | 5 | Jalali et al 2019 |
| Mt Logan NW Canada | 4300±70 | 4000±70 | 2-3 | Fisher et al 2008 |

**Figure 1  4.2 ka BP proxy site onset and terminus**

**Fig. 1.** 4.2 ka BP proxy site onset and terminus

[Figure]

Figure 2 KM-A, Mt Logan, MD04-2276, Gol-e Zard

**Fig. 2.** Mawmluh KM-A, Mt Logan, MD04-2276, Gol-e Zard

---

## Editor Comment (EC3) · Denis-Didier Rousseau (Editor) · 4 Jan 2021

Dear authors,

Reviewer 1 has posted a detailed review that I would encourage you to reply to during the present discussion phase to foster discussion between both of you and therefore prepare the potential revision of your manuscript.

Thank you in advance

denis-didier Rousseau

CP co-editor in chief

---

## Editor Comment (EC4) · Denis-Didier Rousseau (Editor) · 4 Jan 2021

Dear Authors,

Reviewer 2, Harvey Weiss, recently posted his review in which he is strongly criticizing your results and interpretation. I would therefore strongly encourage you to seriously consider his arguments and post a reply that could help deciding on the future status of your paper.

Thank you in advance

All the very best

denis-didier Rousseau

CP co-editor in chief
* * *

---

## Author Comment (AC2) · 11 Jan 2021

We would like to thank the reviewer for their thought-provoking review. They have raised several excellent points, which we have attempted to answer. In doing so, we feel the manuscript has been improved.

"Major points. 1)The MC-PCA analysis is state-of-the-art, but the spatial component of the results go largely unpresented or discussed. This is especially the case for PC2, given the mix of loadings, but even the spatial patterns of the strength of PC1 could

shed light on the question at hand. I suggest the authors add maps that illustrate the PC1 and 2 loadings, and their uncertainties, and discuss their results. This is particularly important for PC2. The authors describe this as a dipole that represents a fluctuating wet and dry signal. The spatial structure of this dipole, especially as it relates the the Harappan civilization, needs to be presented and interpreted; including a discussion of the age uncertainty, and record selection uncertainties (PC-1,-2,-3)."

We agree that the spatial analysis was discussed as thoroughly as it might. The spatial pattern of loadings, particularly PCA-3 PC2, help provide key evidence in our discussion as to why the 4.2-3.9 kyr drought is unlikely to be caused by changes in summer rainfall. We welcome the opportunity to rectify this. We have introduced a new figure (Figure 5) which shows the loadings of PCA-3 PC1 and PC2. In the results section we have changed the description of the loadings section 3 to focus more on the spatial dipole, and (as suggested by the reviewer in a later comment) describing the temporal pattern as areas of wet and dry, rather than describing the whole pattern as wet/dry which was misleading.

The spatial dipole provides compelling additional evidence for a winter drought localized to the Indus valley area. As both the Oman and NW India records plot with the rest of the Indian Ocean rather than the Indus Valley records in PCA-3, PC2, the spatial dipole is unlikely to represent changes in the summer monsoon (which should affect Oman, India and Indus valley together) but rather represents changes in winter rainfall or the degree to which winter rainfall influences the site. We have added to the discussion in section 4.3 to make this argument.

MC-PCA doesn't include its own age uncertainty per se. The age uncertainties of the individual records are accommodated in the y-axis uncertainty of the resulting EOFs. Age uncertainty of individual transitions can be determined independently through a detection algorithm of the 2000 individual principal component analyses, for example using rampfit. We have added a sentence to the methods section to make this clearer.

[Figure]

We have added a sentence to the results to more explicitly state that the similarity in results between the different PCA analyses indicate that record selection is unlikely to play a large role in the inferences drawn.

"2. From the analysis, primarily the gradual decline of the summer monsoon recorded by PC1, the authors propose a "Double-drought" hypothesis, that suggest that the combination of a shorter term (300-yr-long) winter precipitation drought, followed by the monsoon weakening beginning around 3.9 kyr weakening summer precipitation may have driven the collapse. This hypothesis is reasonable, and is supported by the MC-PCA analyses, a synthesis of Harappan archeological sites, and a single speleothem record from Italy (RL4) interpreted to illustrate the decline in winter precipitation in India. The contrast between the first two lines of evidence, and the Mediterranean speleothem is striking, especially given the importance of winter precipitation in the double-drought hypothesis. Unlike the analysis that went into summer precipitation, only a single record (RL4) is shown to draw inference about "Mid-latitude Mediterranean Climate", with a notably different age model than that used in the original (2016) publication. To defend the "Double-Drought" hypothesis, the authors need to better characterize Mediterranean climate, and its relation to winter precipitation on the Indian subcontinent. The paragraph from 359-361 states that this is broadly consistent, but this needs to be robustly established. I recognize that this is a significant request, but it would be ideal to see an analysis, either new or from the literature, of multiple records from Mediterranean (if that is indeed the best way to estimate winter precipitation in India), that handles age uncertainty and disagreement between records. As is, it's difficult to evaluate the hypothesis when this key line of reasoning is so poorly supported."

We agree that this is the obvious next step in the thought process. The argument presented here has a tropical rainfall/Indian Ocean focus. This was by design as we wanted to investigate the tropical climate response to the mid to late Holocene transition rather than get pulled back into mid-latitude climate, which has had numerous

studies on the 4.2 kyr event, including two substantial review papers and a special issue in Climate of the Past in the last three years. However, we recognize that by proposing the Double Drought hypothesis we cannot ignore mid-latitude climate. Our argument perhaps relies too much on proof by elimination: a need to explain the obvious drying in the Indian subcontinent when there is no evidence for summer drought (which is as much as a tropical only study could conclude), rather than firmly establishing the existence of a winter drought outright. We are happy to rectify this. We feel that a detailed study of Mediterranean winter climate is beyond the scope of this paper and would likely not add much beyond the recent Bini et al., 2019 and Kaniewski et al., 2018 review papers. However, a more detailed explanation of the literature is certainly a reasonable request to help characterize Mediterranean winter climate and support our arguments.

We intended RL4 to be illustrative rather than indicative of Mediterranean rainfall. While there are plenty of records from the region showing the 4.2kyr event, there aren't that many that cover 5-3 kyr in the same or better resolution as our PCA-analysis, and therefore able to provide a suitable visual comparison.

A similar MC-PCA analysis to the one conducted in the Indian Ocean, but on the winter rainfall zone across the Middle East would likely not have enough records, mainly because the winter rainfall zone is much smaller than the Indian Ocean, but also because of the challenges of conducting fieldwork in the west of Iran, Afghanistan and Pakistan. A quick review of the stalagmite literature in the Middle East suggests just three or four suitable stalagmite records (including a Borneo style outgroup) and one lake record. Specifically: Jeita cave in Lebanon, which shows a 4.2 in some (Je-1, Je-3) but not all (JeG-stm-1) stalagmites. Sofular cave in Turkey, which shows no obvious signal but is probably too far outside the region and influenced by year-round precipitation. A suitable outgroup record akin to the Borneo record used in our analysis Soreq Cave in Israel is a maybe. Stalagmite 2N terminates growth at 4.4 kyr BP. The multi-stalagmite composite appears to become low resolution at 3.5 kyr BP. Tonnel'naya cave in Uzbekistan is too low resolution and may be more representative of northerly jets to provide a record on the trajectory pathway. Gejkar cave record from Iraq isn't old enough. Mitzpe Shlagim record from Israel/Syria terminates at 4.3 kyr BP. Jerusalem West is too low resolution. Gol-E-Zard record is not long enough. Other non-stalagmite high resolution archives include: The Neor Lake record, which does contain the 4.2 kyr event.

This does not mean there aren't low resolution records that can at least provide an insight as to whether drought propogated across from the Mediterranean, through the Middle East, across Iran and Afghanistan and down through Pakistan to the Indus Valley. And here take the recommendation of the reviewer to provide substantially more evidence. We decided to expand what was formerly one paragraph in section 4.3 on the 'second climate signal' into four paragraphs. The first explains the spatial pattern of the PCA analysis (as suggested in reviewer comment 1), the second on evidence for the 4.2kyr Middle Eastern drought being a winter drought, the third outlines the climate mechanism of Western Disturbances and their importance on winter rainfall variability in north-west India, and the fourth on the paleoclimate evidence for propagation of the winter drought through Iran to the Indus Valley.

We thank the reviewer for this suggestion, as we feel it has made a genuine improvement to our manuscript. As the revised manuscript will likely be submitted later, we include those four paragraphs below to facilitate discussion:

The second climate signal in the Indian subcontinent is an abrupt drying between 4.2 and 3.9 kyr BP with a return to wet conditions afterwards. This signal is observed in our PCA-3 PC2. The spatial dipole of this signal (Figure 5b) with positive loading (dryer between 4.2 and 3.9 kyr) in the three offshore Indus valley records, and negative loading (wetter conditions between 4.2 and 3.9 kyr BP) through most of the rest of the Indian Ocean records. The absence of this signal in PCA-1 PC2 and PCA-2 PC2 suggests the 4.2 and 3.9 kyr BP drying is not a dominant feature of monsoonal rainfall variability in the Indian Ocean basin, although it is locally visible in Madagascar and Australia. Further the Oman and NW India stalagmite records, which are interpreted as proxies

for variability in Indian Summer Monsoon strength via cross Arabian Sea wind strength and integrated moisture rainout respectively (Fleitmann et al., 2007, Kathayat et al., 2017), do not load together to the offshore Indus valley records, as would be expected if the PC2 were a proxy for Indian Summer Monsoon variability. This suggests that PC2, and by extension the 4.2-3.9 kyr BP drying, does not represent Indian Summer Monsoon variability.

Instead, a dry period between 4.2 and 3.9 kyr BP matches the expression of the 4.2 kyr event in the Mediterranean and Middle East as a major drought (Figure 7c). In the Middle East the 4.2kyr event is likely manifested as a reduction in winter rainfall. This is intuitive given the highly seasonal precipitation concentrated in winter months (DJF). Specific seasonally resolved or seasonally sensitive evidence for a reduction in winter rainfall in the region includes a pollen record from Tell Tweini (Kaniewski et al., 2008), a coral record from the Gulf of Oman indicating increased winter shamals and dust storms (Watanabe et al., 2019) and a positive d18O excursion in speleothems from Gol-E-Zard (Carolin et al., 2019). In the modern negative d18Oprecip occurs in Iran during winter months (Mehterian et al., 2017; Carolin et al., 2019).

Winter rainfall over the Middle East is largely sourced from the eastern Mediterranean. The synoptic weather systems continue eastwards in the form of Western Disturbances, with additional moisture from the Caspian and Arabian Seas. Western Disturbances are upper tropospheric cyclonic storms carried by the Subtropical Westerly Jet (Dimri et al., 2015, Midhuna et al. 2020 and references within both), moving across Iran, and Afghanistan into Pakistan and India until they reach the blocking Karakoram and western Himalayas. Western Disturbances are responsible for the majority of the winter rainfall in north-west India and winter snowfall in the western Himalayas (Cannon et al., 2015, Lang and Barros, 2004, Midhuna et al., 2020), and are important moisture sources for growing the winter 'rabi' crops, snowpack and subsequent spring flow of the Indus (Yadav et al., 2012).

Confirming whether a reduction in winter precipitation in the Levant and Mesopotamia

at the 4.2 kyr event propagated through Iraq, Afghanistan and Pakistan to the Indus valley requires paleoclimate records from along the moisture trajectory. Insufficient high-resolution records yet exist (Burstyn et al., 2019) to provide a comparable analysis to the Indian Ocean synthesis in this paper but in general, available high-resolution records tend to show a dry event beginning at 4.2 kyr BP while lower resolution records show a mixed response. From west to east the current evidence includes: a multi-proxy record from Mirabad Lake in Iran (33.08°N 47.71°E) which shows no severe drought (Stevens et al., 2006), but may not have the sampling resolution to see such an event. The Neor Lake (37.96°N, 48.55°E) record of aeolian input and hydrological conditions shows increased dustiness (Sharifi et al., 2015). The Gol-E-Zard speleothem d18O records (35.84°N, 52.00°E) (Carolin et al., 2019) indicates a reduction in winter rainfall. Pollen analysis from Maharlou Lake (∼29.45°N, 52.75°E) suggests no major upheaval in the mid-late Holocene transition, with continuous human cultivation (Djamali et al., 2019). A lake sediment magnetic susceptibility record from Lake Hamoun in Iran (30.93°N, 61.25°E) suggests some kind of transient dry event around the middle to late Holocene, but the dating is insufficient to confirm a 4.2 kyr BP timing (Ali Hamzeh et al., 2016). In the Indus valley itself, carbon isotope values of rice grains, a summer crop, show increased drought only after 4.0 kyr BP in both Gujarat and Indus valley locations, indicative of no substantial change to summer rainfall between 4.2 and 3.9 kyr BP (Kaushal et al., 2019).

"3. In new studies that rely on syntheses of multiple records, it's critical that the data (and ideally the code) used to conduct the analyses is available, and replicable. Data that are only available upon request are not publicly accessible, and I strongly encourage the authors to archive the data used in the analysis in a public data repository. If they cannot, the authors need to explain why they cannot follow the best practice recommendation in the "Data availability" section." We agree 100%. 1) PCA data: as stated in our Data Availability statement: "at the NOAA Paleoclimatology Database: https://www.ncdc.noaa.gov/paleo/study/xxxxx." with the exact url updated at publication 2) New data: The new stalagmite record from Madagascar will be archived in

the NOAA database and submitted to the SISAL database, as per the data availability statement of the companion submission (cp-2020-137). 3) Previously published data: Supplementary Table 2 lists the SISAL entity ID for the six stalagmite records discussed in the text, along with dois for the marine cores with publicly available data. It is not our place to share data already publicly archived that was not generated by us. 4) The analytical code in this study is written by others and properly cited: namely Deininger 2017 and Mudelsee 2000 and 2013. It would be a breach of copyright to share this code. Our statement on making the data available upon request is an additional courtesy extended to all researchers who might wish to contact the authors to discuss and use the data. We agree entirely that data should also be publicly archived, and it will be, as per the data availability statement.

"4. Why is PC-3 truncated? Looking at the data in figure 2, the three sedimentary records span the full length of PC-1, and PC-2. This decision needs to be explained and justified." Truncation of the records is part of the Deininger software. First, the software does not extrapolate beyond individual ages to the top and bottom of archives, so the records used will always be shorter than those published. Second, the PCA has to have 2000 full realizations of all included records. Therefore, the younger limits are determined by the 1 sigma older bound of the youngest age of the earliest finishing record. In the case of PCA1 and 2 this is the Rodrigues stalagmite record at 3070 yrs BP. The older limit is determined by the 1 sigma younger bound of the oldest age of the latest starting record. In the case of PCA1 this is around 5600 yrs BP so not an issue which influences the 5000-year cutoff. In the case of PCA2, the Madagascar record provides the limit at 4950 yr BP.

For PCA3 the issue is slightly different. The Deininger software requires not just a mean resolution below the threshold, but a data-point in every bin. For this reason, the Lake Rara Mn/Ti and Flores stalagmite records were not included in the analysis as they contained areas of higher and lower resolution proxy measurements. The Giesche record does contain a data-point in every bin. But only for certain age realizations. The

stretching and compression of the age model that occurs in each realization occasion-ally pulls data points out of a bin, leading to a truncation of the PCA time period. This is particularly important for the radiocarbon ages in the Giesche record, where the error bars at 4600 yr BP are +-100 years, compared to +-40 years at other depths, leading to much larger stretching of the record.

PCA3 is therefore a balance between length of record, and resolution of record. A resolution of 20 years necessitates truncating the record at ∼4400 years. The same is true up to 35year resolution. There are slight improvements with decreasing reso-lution: 4450 at 40 years, 4500 at 45 years, and 4650 at 50 years. This is ultimately a judgement call between length of analysis and resolution of analysis. As this study is designed to use only the highest resolution records, we decided that the 4.5 to 3.3 time period provided by PCA-3 covers the major periods of interest.

We have added a description of this to the methods section so that the decision is at least explained and justified. We have kept the analysis as is, but are willing to change PCA-3 to a 50 year resolution if required.

"Detailed points: 38-39: Awkward to say "increasingly recognized", and then have only references from 2003 and 2004." Oops, yes. We have included a 2011 reference, but there are few non-local overview papers from the last five years, so we also removed the increasingly.

"115: "Between the records"? Do you mean between the ages?" Yes, changed as suggested. Thank-you

"116: Upscaled. Might be better to say "degraded"." We disagree. Upscaled is equally precise, but less emotive. Upscaling and downscaling is also the more commonly used term e.g. Anchukaitis and Tierney 2012, Deininger et al., 2017)

"140-144: Why? It seems easy enough to use identical ensemble members each time through. " We have tried to minimize the number of changes made to the Deininger

code. Changes to the code remove the reproducibility of our work. Using identical ensemble members does not change the outcome of this paper, and our use of different age models gives more conservative error bars rather than less, so we are not overinterpreting the results.

"More importantly, can you justify treating multiple records from the same core as independent climate records?" This is a good question. It seems inevitable that different foram species d18O at the same location have a certain degree of correlation that is independent of the climatic variable attributed to each one individually, and therefore the records are not truly independent. We note that the original Giesche paper interprets winter rainfall as resulting from changes in three foram species via the delat of both sacculifer-ruber and duterrei-ruber proxies while interpreting summer rainfall from one of the foram species (ruber). The two seasons are therefore not completely independent and could explain why Giesche concludes a simultaneous summer and winter drought. Our treatment of the two records (G. ruber and N. dutertrei) rather than the interpretations (Summer and Winter rainfall) removes as much of the interdependence as possible. The addition of the Arabian Sea record does go someway to alleviate the problem by providing a third record from the area, and one that has a similar positive PC2 Loading. Further, the close agreement of PCA3-PC1 which includes all three Arabian Sea records with PCA1-PC1 and PCA2-PC1 which include none suggest that any codependence of two records makes little difference to the overall outcome of the paper.

"172: I suggest you start by describing this as a dipole, and then describe it as fluctuating wet and dry. 178: I don't think you can call this a dry period in PC2, since it's a dipole, it must be dry some places and wet others. In general, the spatial characteristics of PC2 should be fleshed out." We agree with both of these comments. We have restructured the paragraphs to talk about in terms of spatial variability of possible dipoles rather than wet and dry conditions. We also agree that the spatial characteristics of PC2, especially from PCA-3, require additional discussion. While we have

added a little here, we feel it is best to interpret the spatial dipole in the discussion. We have added several sentences to the paragraph starting line 260 in section 4.3. Thank-you.

"294: 4.26 to 3.97 kyr BP this is a lot of precision, where do these numbers come from, and what are their uncertainties? This is comment relevant throughout the discussion." 4.26 and 3.97 in regard to the winter drying both come from Carolin et al. 2019 as cited in their first useage on line 41. We have chosen to report this number to same number of significant figures as the original paper.

3.97 and 3.71 in regard to the summer drying comes from the rampfit analysis from PC1. The standard error on these number are 92 and 95 years. We have added a quick explanation of where these numbers come from to line 296 to make this clearer.

In general, a good rule of thumb for error reporting is two significant digits on the error, and the same number of decimal places for the value. In theory this allows us (just) to report the rampfit derived values to the nearest year. However, given that the quoted errors are one standard error, we can see why this might be viewed as a stretch. Therefore, we have changed all mentions of these values to the nearest ten years (one significant digit of error). We feel this is a reasonable degree of accuracy that does not overstretch these numbers.

At all other points in the manuscript, we quote ages to the nearest 100 years which is comparable to radiocarbon error at this age, with the occasional use of the nearest ten years when defined as such in the original literature (e.g. line 240).

"309: What does this analysis say about whether the mature Harappan occurred during a short term pluvial?" Our data supports this idea. We do have a sentence later in the manuscript that outlines this. However, it is clear this reasoning should be presented earlier. Therefore, we have added a sentence to this paragraph. In particular, PCA3 PC2 shows wet conditions between 4.5 and 4.3 kyr BP. It is also likely that both PC1 and PC2 from PCA1 and 3 show a modest and substantial increases in moisture

between 4.7 and 4.5 kyr BP respectively. But as these analyses say less about rainfall in the Indus valley stating such could be an overinterpretation of the data.

"314: "was likely caused by a reduction in western disturbances that bring rainfall from the Mediterranean and Middle East (Figure 6c). "How do you know this?" This was said with far more certainty than it should have been. The idea of reduced winter rainfall is hypothesized by us to explain a reduction in rainfall that our analysis shows was unlikely to be caused by a reduction in summer rainfall. That this reduction in rainfall was caused by reduced western disturbances is an inference from the idea from Giosan et al., 2018 that the increase in winter rainfall three hundred years previously was caused by an increase in western disturbances. We hope that our new wording of this paragraph makes this clearer. See response to Q2 for more detail.

"Figure 1. Add Harappan sites" Changed as suggested

"Figure 5. Is this the ratio of winter to summer? Or the fraction that winter represents out of total rainfall. " This is the ratio. ie. winter (DJAF) divided by summer (JJAS). We have amended the figure axis label and caption to make this clearer.

"Also, an overview map showing where this is would be helpful for those less familiar with the region " Changed as suggested

"Figure 7. The d) label is a little hard to locate. Consider moving it to the top left corner." Changed as suggested
* * *
a)

63KA, Indus Fan
G. ruber
N. dutertrei
Qunf Cave
Oman
SO1320-289KL
Arabian Sea
Sahiya Cave, NW India
Bukit Assam Cave
Borneo
Tangga Cave
Sumatra
Anjohikely
Madagascar
La Vierge
Rodrigues
KNI-51
Australia

PC1 Loading
-0.55   0   0.55
■ Stalagmite   ● Sediment Core

b)

63KA, Indus Fan
G. ruber
N. dutertrei
Qunf Cave
Oman
SO1320-289KL
Arabian Sea
Sahiya Cave, NW India
Bukit Assam Cave
Borneo
Tangga Cave
Sumatra
Anjohikely
Madagascar
La Vierge
Rodrigues
KNI-51
Australia

PC2 Loading
-0.55   0   0.55
■ Stalagmite   ● Sediment Core

**Fig. 1.**

[Figure]

---

## Author Comment (AC3) · 29 Jan 2021

In their review, Professor Weiss lists numerous regional and global hydroclimate anomalies that overlap, within the age uncertainty of those records, the 4.2 kyr event, and presents them as evidence of a synchronous climate event regardless of the seasonality of the proxy or the shape of the anomaly. Our aim is not to deny that these hydrological anomalies exist. We dedicate section 4.2 of our manuscript to describing these anomalies as they occur on the Indian subcontinent. Rather our aim is to put them into regional context. The seasonality of any individual paleoclimate proxy

is likely ambiguous. As the paleoclimate community moves towards integrating data from multiple sources ("big data") and accounting for time uncertainty, the original interpretations of individual records may not all hold in light of improved understandings of regional spatial variability.

By studying numerous records across the Indian subcontinent, we are able to separate out these records into three separate groups – 1) gradual drying, 2) 4.2kyr event abrupt drying and reversal at 3.9 kyr BP, and 3) a step-change drying at 3.97 kyr BP. The first of these is a well-recognized millennial scale trend over the entire Holocene. The second is consistent with drying seen across the Middle East (now discussed in more detail thanks to the suggestions of reviewer #1). The third is consistent with regional Indian Ocean dominated records as indicated by our PCA analysis. The low-resolution paleoclimate records are all real records, recording real changes in the hydrological system, and they all fit one or more of the three drying signals. As we argue in our manuscript in the final paragraph of section 4.2, "they cannot all be recording regional variability in the summer monsoon". Therefore, it is impossible to interpret every single record as originally interpreted in the original manuscripts, and impossible to interpret every single record as a proxy for Indian Summer Monsoon rainfall amount. Further, the age uncertainty of many records may make it difficult to distinguish between these different drying events.

The purpose of our PCA analysis is to use only the highest resolution records to tease out the different climate signals contributing to regional hydroclimate variability. The advantage of using high resolution, precisely dated, paleoclimate data, is to provide higher resolution information on past climate, i.e., to separate apart multiple climate signals that may be closely spaced in time and not distinguishable in lower resolution or poorly dated records. Our results do not refute any of these records as real indicators of hydrological change. Simply, our results provide a framework in which lower resolution records can be interpreted in light of the multiple climate anomalies that occur within a short space of time over the mid- to late- Holocene transition.

In comment #4, Professor Weiss claims that we change the chronology and spatial pattern of abandonment of the Harappan civilization. We have not done so, and the chronology and spatial pattern of abandonment follow the literature. The robustness of the radiocarbon data is certainly up for debate, and we recognize that ongoing and future work will surely refine the exact spatial and temporal pattern of the rise, decline and urban abandonment of the Harappan civilization. However, the chronology as presented in the literature cannot be robust enough to allow for a 4.2kyr event summer drought, or a 4.2kyr event combined summer and winter drought as proposed by others, yet simultaneously be insufficiently robust to allow for a 4.2kyr event winter dominated drought. Our data rejects the idea that Harappan decline was forced predominantly by variability in the summer monsoon.

Overall, we find that this review argues that we must be wrong because other records are interpreted differently. The review provides no compelling argument against the methodology, the results, or the idea of three separate dry events in the region, each one derived from climatic changes in different moisture source regions and on different timescales. The arguments presented here against the Double Drought hypothesis rely on specific interpretations of hydrological records. As we explain below in detail, it is not possible to interpret every hydrological record in the circum-Indian Ocean basin as being controlled by the Indian Summer Monsoon. Deconvolving different modes of climatic variability that occur synchronously or within a short space of time is an important step in understanding mechanisms that drive our climate. The Double Drought hypothesis does this. Our interpretation of the spatial pattern of these three separate climate anomalies assigns reasonable climate mechanisms to each climate anomaly. We assign winter rainfall variability to climate dynamics in the source region of winter rainfall, and summer rainfall variability to climate dynamics in the source region of summer rainfall. The Double Drought hypothesis builds on the work of previous paleoclimate studies. It is consistent with regional climate dynamics, regional hydroclimate data, archaeological data, and archaeobotanical data.

FIRST COMMENT: "The IUGS-recognized global boundary stratotype for the 4.2 - 3.9 ka BP event, marking the middle to late Holocene transition to the Meghalayan stage, is the KM-A speleothem $\delta$18O record from Mawmluh Cave, Meghalaya, NW India, that is an Indian Summer Monsoon (ISM) drought record (Berkelhammer et al 2012; Walker et al 2019)." The assumption that Mawmluh Cave KM-A record is a 100% Indian Summer Monsoon record has recently been challenged. A detailed sub-seasonal trace element (Ronay et al., 2019) record from Mawmluh Cave indicated that winter rainfall variability has a significant, if not dominant, influence on proxy variability. The authors explanation is that summer rainfall is so substantial at the cave site that variability in summer rainfall does not alter annual proxy values by a significant amount. Instead, as winter rainfall is much more variable, this variability contributes much more significantly to annual proxy variability. These results were corroborated by the d18O record. We find the work of Ronay et al reasonable and intuitive. We have added the Ronay reference to our manuscript.

Indeed, if the KM-A record is interpreted as responding to both summer and winter rainfall then our results and interpretation match the KM-A record very well. KM-A contains two step changes in d18O which both simultaneously define the 4.2 kyr event of the GSSP. Both step changes in the KM-A record are supported (within a 30-50 year age model adjustment/error) by the two hydrological changes implicated in our study (a winter change at 4.26 and a summer change at 3.97).

"The recent analysis of the Indus delta foraminifera record at core 63KA has identified, as well, the Indian Winter Monsoon drought synchronous with the 4.2 ka BP ISM drought (Giesche et al 2019). The global boundary sub-stratotype is the Mt. Logan Yukon glacial core's $\delta$18O moisture event (Fisher et al 2008). Scroxton et al present a principal components analysis of seven recent $\delta$18O speleothem records from the Indian Ocean region and the Giesche et al 2019 delta foraminifera analyses (line 110) "to investigate the impacts of the 4.2 kyr event on tropical Indian Ocean basin monsoonal rainfall" and the late third millennium BC Indus urban collapses. Similar to

earlier analyses using lake sediment records (e.g., Leipe et al 2014), Scroxton et al note the succession of two gradual centuries long dry periods separated by the 4.2 ka BP aridification event, but from their analysis present four new conclusions."

Our manuscript does not present two gradual centuries long dry periods separated by a 4.2 kyr event. Our manuscript presents two consecutive centuries long dry periods, one of which is the 4.2 kyr event. This misrepresentation probably comes from the Leipe record, which does show two droughts separated by a brief (2 data point) half return to wetter conditions.

"Scroxton et al conclude that the Mawmluh Cave KM-A speleothem is not a useful stratotype because (a) line 370 "The KM-A record replicates neither the other speleothem from Mawmluh Cave (Kathayat et al., 2018). "As previously noted, Mawmluh KM-A is similar within standard deviations to the other Mawmluh speleothems ML.1, 2 in both onset and terminus (Kathayat et al 2018)."

In our opinion, the term "similar within standard deviations" is ambiguous. Is this referring to temporal replication, isotopic values, normalized isotopic values or isotopic change? In figure 2 of our manuscript, we present a series of local and regional hydroclimate records including the Mawmluh Cave KM-A record that support our statements. The Mawmluh cave record is defined principally by positive isotope excursions, one at 4.30 kyr BP, and one at 4.05 kyr BP. The Mawmluh Cave ML.1 record contains neither excursion. Most likely, it contains a step-change around 4.0 kyr BP on a gradual drying trend – the two other regional climate anomalies described in the paper. ML.2 largely replicates ML.1, especially in the gradual trend, though the step change is less obvious. Is there evidence of a 300 year long, abrupt, transient 4.2 kyr BP event that represents a significant climatic departure from normal? In our opinion this event is not obvious. There is minor variability in both records around 4.25 kyr BP – both stalagmites show abrupt isotopic excursions. However, the excursions are of opposite sign between the two stalagmites, and the duration of the excursion is only 20-30 years. We stand by our conclusion that KM-A is not replicated in its own locality.

"nor any regional records (this study)." "It is similar, as well, to the records at the Indus delta (Giesche et al 2019), and to the sampling resolutions of the two recent ISM Madagascar speleothems (Wang et al 2019; Scroxton et al 2020 in review). Not listed here is the Sahiyah Cave, NW India speleothem that certainly does not present an abrupt 4.2 ka BP event, but was "manifest as an interval of declining ISM strength, marked by relatively higher amplitude of del 18O variability and slow speleothem growth"(Kathayat et al 2017). Mawmluh KM-A is also congruent and synchronous with the high resolution speleothem westerlies proxy for the 4.2 ka BP dust/drought event at Gol-e Zard NW Iran (Carolin et al 2019) as shown in Figure 2 top, that is synchronous with the settlement collapse in northern Mesopotamia (Weiss et al 2012) and many regional settlement abandonments across the Mediterranean."

The 4.2 kyr event cannot be all climatic variability between 4.5 and 3.5 kyr BP. In the Mediterranean and Middle East, where the evidence is strongest and most numerous, the 4.2 kyr event is 1) abrupt, 2) transient (an excursion rather than a step-change), 3) occurs between 4.25 and 3.95 kyr BP. We disagree that widely observed step-changes in climate at 4.0 kyr BP are manifestations of the 4.2 kyr event. We disagree that gradual secular trends over 1000 years are manifestations of the 4.2 kyr event. We agree that there are similarities between the KM-A record and other regional records. Indeed, as outline above, we suspect that the KM-A stalagmite may record both winter and summer rainfall variability and therefore both of the drying events. We do not agree that the Sahiya cave replicates an abrupt, transient 4.2kyr event because it shows gradual isotopic variability between 4.5 and 3.5 kyr BP. The Gol-E-Zard record in Iran is not a record of summer monsoonal rainfall, rather it likely records winter rainfall from westerly derived moisture in the d18O record, and regional dustiness in the Mg/Ca record. The two Madagascar records are not records of Indian Summer Monsoon rainfall, numerous paleoclimate studies indicate that southern hemisphere monsoon systems can act in phase, out of phase or without phase to the northern hemisphere under different climatic conditions. Settlement collapse in the Middle East is likely not dependent on summer monsoon rainfall and is certainly not dependent on Indian

Summer Monsoon rainfall.

The purpose of this study and manuscript, and the increasing number of big-data climate papers being produced, is to analyze the results of hydroclimate proxies independent of their original interpretations of seasonality, but rather in the context of regional variability. The similarity of one record to another is not indicative of both being caused by variability in the Indian Summer Monsoon.

We understand how "nor any regional records" might be interpreted as implying a lack of replication with any record, anywhere, rather than the outcome of this study as implied by the phrase "this study". We should have been more careful with our wording. We have decided to update this section so as to be more precise. However, our interpretation and conclusions have not changed.

"and (b) line 374 "is low resolution"

We agree that KM-A does not have low sampling resolution when compared to other stalagmite records from the region. We have removed this statement.

"low dating frequency," The KM-A record has three U-Th datapoints between 5100 and 3600 yr BP. An average spacing of 750 years. The ML.1 record has 18 U-Th ages between 3.7 and 4.5 kyr BP (44 year average), and the ML.2 record has 3 (266 years). Between 3.0 and 5.0 kyr BP: the Oman record has 6 U-Th ages (333 years). The Sahiya record 15 (133 years). Our Madagascar record has 7 (285 years). The Rodrigues record has 16 (125 years). The Australian record has 15 (125 years). The Borneo record has 6 (333 years). The sediment core record of Giesche has 7 (285 years). We stand by our statement that the KM-A record has a low dating frequency.

"not replicable within its own locality," The response to replication at Mawmluh cave is answered above.

"ambiguously defining a climate event" We stand by our statement that the definition of the 4.2 kyr event is ambiguous. We attach a figure showing the KM-A record in detail.

The 4.2 kyr event is defined as being midway between two events: "The first registration of the event in the stable isotope record occurs at ∼4300 yr BP followed by a second marked increase in stable isotope values at ∼4100 yr BP. The abrupt increase in stable isotope values is the primary boundary marker for the GSSP, and hence a date of 4200 yr BP, which effectively marks the mid-point between these two modelled ages" (Walker et al., 2018). The GSSP is therefore defined by a point between two stable isotopes (red dots) during a period of twenty consecutive stable isotope points with less than 1.01 per mill variability. Our analysis demonstrates that the two anomalies are likely caused by separate climate events.

"that is not significant across its climate domain". We understand that "not significant across its climate domain" may be not be as precisely worded as it should be. However, we have demonstrated above that the KM-A record is not representative of summer monsoon rainfall, its interpreted climate domain.

Overall, we are happy to update our manuscript, to be more precise with our phrasing, and to spell out more clearly our reasoning behind our statements. The updated wording of section 4.5 is:

"The Mawmluh Cave speleothem records which define the 4.2 kyr event (Berkelhammer et al., 2012; Walker et al., 2018) are too short to be included in our PCA analysis. However, the highest resolution replicated record from the cave, ML.1, replicated by ML.2, shows gradual drying over its entire growth period, wetter than normal conditions at 4.1–4.0 kyr BP, and a step-change increase in ïĄd'18O at 4.0 kyr BP (Kathayat et al., 2018). These results are consistent with both the secular millennial scale drying trend, and the 4.0 kyr BP summer monsoon drought identified in PC1. The Global Boundary Stratotype Section and Point (GSSP) golden spike is located in stalagmite KM-A (Berkelhammer et al., 2012). Stalagmite KM-A does not replicate ML.1 or ML.2 from the same cave. Our results indicate that KM-A is not representative of hydroclimate variability in the Indian Summer Monsoon domain. Instead, KM-A contains two increases in ïĄd'18O (drying events), one at 4.31 kyr BP and one at 4.05 kyr BP. Hydroclimate proxy variability at this site may be significantly influenced by dry season variability {Ronay et al., 2019, #72351}, suggesting KM-A may not be an exclusive record of summer monsoon rainfall. Within reasonable age uncertainty (±30 years at the nearest U-Th date, so likely slightly higher away from the age) the timing of the KM-A dry anomalies beginning 4.31 kyr BP and one at 4.05 kyr BP are consistent with both 4.26 and a 3.97 kyr BP drying events. We hypothesize both a winter and summer influence stalagmite ïĄď18O in KM-A.

There are also notable issues with the stalagmite itself. While the ages of KM-A are very precise, there are only three ages between 5084 and 3654 yr BP and stalagmite growth is very slow. The shape of the stalagmite after 5 kyr BP is not convincing of unaltered equilibrium deposition. Even if the ïĄď18O record of KM-A does record both drying events, the golden spike location is defined as 4.20 kyr BP, part of a run of 40 consecutive ïĄď18O samples between the two drying events, with relatively minor variability (<1‰. The existence of the Northgrippian-Meghalayan GSSP golden spike in a low dating frequency record, not replicable within its own locality, not representative of climate variability across its climatic domain, and ambiguously defined as the mid-point between two different climate events, is problematic at a minimum (Helama and Oinonen, 2019). We recommend that an alternative GSSP golden spike for the mid- to late-Holocene transition be identified."

SECOND COMMENT: "Secondly, Scroxton et al argue that the 4.2 ka BP event was of (line 400) "limited impact on tropical monsoonal rainfall around the circum-Indian Ocean basin" Where we have missed records, we are happy to include them and have incorporated some of the suggested studies into our paper. But as of yet, none of the suggested records dispute the three drying events of our hypothesis: 1) the secular trend, 2) a 4.25 kyr transient event and 3) a 3.97 kyr BP step-change.

However, Scroxton et al do not include the Mawmluh KM-A speleothem record in their principal components analysis (line 364, "too short to be included") Principal component analysis requires records covering the entire duration of the interval of interest. In

not covering the 5-3kyr interval, the KM-A, ML.1 and ML.2 records are too short to be included. We discuss lower resolution records extensively in section 4.2, and dedicate section 4.5 to an in-depth discussion of how the three Mawmluh cave records fit within our new framework. In summary the three Mawmluh cave records show, between them, all three drying events.

"nor the ISM Tibetan plateau speleothem record (Cai et al 2012)," The Cai et al., 2012 record stops growing at 4.15 kyr BP (+- 180 years). It therefore cannot be used in the PCA analysis. A hiatus at this point is indicative of some kind of drought, and is within error of both drying events at 4.26 and 3.97 kyr BP. We have included this record in section 4.2.

"and the possibly anti-phase Southeast African records (Humphries et al 2020)." We discuss both the 2019 and 2020 Humphries records from southern Africa in detail in our companion manuscript. They are not included in this compilation as they are not summer monsoon records deriving moisture from the ITCZ. Instead, they derive the majority of their moisture from the SE Trades, with some influence from tropical temperate troughs, which have their own complex tropical/mid-latitude climatic controls.

"Most problematically, however, Scroxton et al ignore the Horn of Africa, the Ethiopian highlands, as there are no speleothem paleoclimate records. Nevertheless, the Ethiopian highlands are a major component, multiply recorded, of ISM sourced Indian Ocean basin hydroclimate." We have not included the Horn of Africa in our detailed analysis because 1) there are no records of sufficient resolution to include in our PCA, 2) we have reserved discussion of smaller sub-regions to a) the Indian Summer Monsoon region with direct relevance to the Harappan, and b) the South-East African Monsoon region as discussed in the companion manuscript. This is in part because 3) the Horn of Africa double monsoon season is not the same as the Indian Summer Monsoon, having both different wet seasons and a different sensitivity to zonal variability (the Indian Ocean Dipole). 4) The progression of climate change in the Horn of Africa during the Holocene is dominated by the African Humid Period, drying in the
mid- to late- Holocene may therefore have little dependence of the 4.2 kyr event. The Horn of Africa is therefore currently out of scope, as is the Old Kingdom of Egypt. This would be an entirely different paper.

Nevertheless, we are happy to discuss individual records here as it is indeed probable that some climatic processes may be similar. We also include a second figure here that demonstrates some of the East African records. We do not believe the discussion below is within the scope of this paper.

"The 1200 mm of highland Ethiopian ISM precipitation collect at Lake Tana, and become the Blue Nile and Atbara Rivers, which together provide 90% per cent of Nile peak flow as measured by air mass back trajectories and 97% of Nile annual sediment load (Williams 2019:28; Woodward et al 2014; Costa 2014). The Nile River extends 4759 kms from Lake Tana to the delta (William 2019: 117), or 2000 kms longer than the Indus, and was the primary physical determinant of ancient Egyptian irrigation agriculture. The sediment core from Lake Tana documents an important, albeit 200 year resolution, low stand at 4.2 ka BP (Marshall et al 2011)" The Marshall et al., 2011 Lake Tana Ti record does indeed show an excursion around 4.2 kyr BP with an age uncertainty of 150 years. However, the lake Tana dD record (Costa et al., 2014) shows wet conditions between 4.6 and 4.0 kyr BP and a single dry data point at 3.7 kyr BP, before returning to dry conditions.

"synchronous with other East African records that include the Lake Mega-Chad large scale dust mobilization (Kröpelin et al 2008; Francus et al 2013)." The Lake Yoa record of Kropelin 2008 shows a step change between 4.2 and 3.9 kyr BP, this is not an abrupt 300 yearlong 4.2kyr event anomaly. The Lake Yoa record of Francus 2013 has a separately defined stratigraphic unit between 4.3 and 3.9 kyr BP but is described as a gradual transition between the units above and below. We interpret neither as showing an unambiguous abrupt anomalous event.

"This ISM/Nile source reduction, known at the Nile delta within numerous and various
sediment core proxies, is recorded at the Nile deep sea fan marine cores. For example, the recent analyses of MD04-2276 include the 4.2 ka BP SST (Jalali et al 2017)" The Jalali et al MD04-2726 record shows a transition in SSTs around 3.8 kyr BP. A longterm of increasing SSTS begins at since 6kyr BP, peaking around 3.9kyr BP before a decrease in SSTs gradually over 400 years to a minimum around 3.5 kyr BP. The paper also describes a more humid interval from 4200-3000 yr BP. We interpret this result as showing two of the three climate anomalies: the long-term secular trend and a step-change sometime around or just after 4.0 kyr BP, but not an abrupt transient 4.2 kyr event. There is a two data point excursion between 4.1 and 3.9 kyr BP. It is of the same magnitude as another excursion at 4.4 and very similar in magnitude (just starting from a different baseline) at 4.55 and 4.9 kyr BP. If this anomaly is the 4.2kyr BP event, it is not of unusual magnitude, and merely represents normal centennial scale variability on top of millennial scale long term change.

"and Mn/Al flux (Mologni et al 2020) events. Although there is interpolation across three radiocarbon dates, Figure 2 middle displays this SST event synchronous and congruent with the Mawmluh KM-A record." The Mologni core describes in detail conditions between 10,200 and 7,200 kyr BP. Above which the core is heavily bioturbated. There are two ages in the bioturbated region of 5.8 and 3.9 kyr BP defining this section. There is a peak in Mn/Al flux and minima in the log(Ti/Ca) in the section. The data is not publicly available to provide quantification here, but the anomalies appear to be roughly 1500 years long.

Other records in the East African region include the Lake Turkana TEX86 and BIT index records of Lake Turkana (Berke et al., 2012), and the Lake Victoria Diatom PCA2record of Stager et 2002, and the Pilkington Bay (also Lake Victoria) Diatom PCA2 record of Stager et al., 2003. None of these records show an abrupt, climatic event at 4.2 kyr BP. Records south of the equator in East Africa are discussed at length in the companion paper to this manuscript.

The 4.2 kyr event is visible in the Gulf of Oman (Cullen et al., 2000) as a winter dust

record but could reflect aridity in any season. In the Arabian sea low resolution records without the 4.2 kyr event include the Arabian Sea G. bulloides record of Gupta et al., 2003, the Gulf of Aden Leaf Was dD record of Tierney et al., 2013 and the Arabian Sea sediment lightness record of Schulz et al., 1998. This represents an exhaustive search of all records with at least 7 data points between 5000 and 3000 years with publicly available data in the NOAA repository.

The significance of the 4.2 ka BP Nile event resides in its synchronism with the Old Kingdom collapse and the beginning of the First Intermediate Period (Barta 2019), one feature of which was considerable settlement abandonment at the Nile delta and resettlement in Middle Egypt. While the Old Kingdom collapse was also synchronous with rain-fed settlement abandonment across Mesopotamia and Syria at 2200 BC (Ramsey et al 2010; Weiss et al 2012), Upper Egypt and its Kerma culture, close to the source of Nile flow, did not experience similar Nile flow reductions nor regional settlement abandonments (Woodward et al 2014). The influence on the Old Kingdom of Egypt is not within the scope of this paper. We do not mention it in our manuscript.

THIRD COMMENT: Thirdly, Scroxton et al state, (line 47) that "the areal extent of the 4.2 kyr BP event beyond the data-rich heartland of Mediterranean Europe (Bini et al 2019) and Mesopotamia (sic) (Kaniewski et al 2018) is unclear." But the event records are abundantly available. The 4.2 ka BP event extended to Australia (Deniston et al 2013) and to southern and Northern China, an ISM and East Asian Summer Monsoon (EASM) event, e.g., Hulun Lake (Zhang et al 2020), Dongshiya Cave (Zhang et al 2018), Lake Balikun (An et al 2011). At Lake Wuya in North China (Tan et al 2020) the event is recently described erroneously as gradual when $\delta$18O increased and decreased abruptly at 4200 and 3996 ka BP. The EASM is, of course, ISM sourced (Liu et al 2015; Yang et al 2014), and a North Atlantic wavetrain for 50% of modern ISM drought events has now been identified (Buhar et al 2020).

Along the western Pacific the event is recorded in Japan (Park et al 2019) and in the Kuroshio Current's "Pulleniatina minimum event" (Zhen et al 2016; Zhang et al 2019;

Shuhuan et al 2021) where its northeastern trajectory likely generated the Mount Logan Yukon 4.2 ka BP event (Fisher et al 2008). The Mount Logan 4.2 ka BP sub-stratotype, with 2-3 year resolution, is both synchronous and congruent with the Mawmluh KM-A event (Figure 1, Figure 2 bottom).

Across mid-latitude North America the event is also well documented, from western Idaho to western Massachusetts, "with median moisture levels reaching a minimum from 4.2 to 3.9 ka" (Shuman and Marsicek 2016: 42; Shuman et al 2019) alongside the North American monsoon phase change recorded at Leviathan Cave (Lachniet et al 2020). In the South American Monsoon region, along the western coast of South America, northern sediment cores suggest abrupt wet eastern Cordillera events synchronous with abrupt dry western Cordillera and Altiplano events. Lake Titicaca, for example, experienced an abrupt diatom shift at ca. 4300 BP followed by a drought event from ca. 4200 - 3900 BP with a lake level drop of ca. 70 meters (Weide et al 2017). At the southernmost Andes, "Marcel Arevalo" caves MA 1-3 record a uniformly wet period from ca. 4.5 to 3.5 ka BP interrupted by an abrupt ca. 23% drop in precipitation centered at ca. 4.2 ka BP (Schimpf et al 2011), possibly associated with several volcanic eruptions.

Synchronously, the continental monsoon along Brazil's east coast and the South Atlantic Convergence Zone that crosses Brazil, experienced abrupt and radical alteration. The Lapa Grande speleothem's sharp, decreased spike in $\delta$18O extended from ca. 4.2 – 3.9 ka BP (Stríkis et al 2011). At Chapada do Apodi, Northeastern Brazil, high resolution speleothems, clastic sediments and bat guano analyses display abrupt high $\delta$13C and $\delta$18O and low 87Sr/86Sr values indicating "a massive episode of soil erosion. .the beginning of the Meghalayan chronozone, characterized as the aridification of this region, decline in soil production, drying out of underground drainages" (Utida et al 2020).

Apart from the dense distribution of Mediterranean and Western Asia records, including the Gol-e Zard speleothem congruent with Mawmluh KM-A, Figure 2 top, the 4.2 ka BP

event synchronous records extend to Alpine Europe (e.g., Spannagel Cave, Fohlmeister et al 2012) and more than fifty subpolar North Atlantic records (Weiss 2019). The latter are now complemented by the high resolution north Iceland marine core MD99-2275 SPG event dated 4290±40 ka BP (Jalali et al 2019; Figure 1) and the Irminger Current event (McCave and Andrews 2019), which both suggest a 4.2 ka BP AMOC slowdown. This was due, possibly, to the freshwater dosing associated with glacial melt documented synchronously, for example at the Agassiz glacial core (Vinther et al 2009; Fisher et al 2012; Lecavalier et al 2017).

We recognize that the phrase "the areal extent of the 4.2 kyr BP beyond the data-rich heartland of Mediterranean Europe (Bini et al 2019) and Mesopotamia (Kaniewski et al 2018) is unclear" may be considered an overstatement. Through numerous discussions over the past few years of this project we believe it to be an accurate portrayal of the opinion of the paleoclimate community. As there is no systematic, detailed review of every single global record spanning the 4.2 kyr event it is not a citable fact. We are happy to change the phrasing of this sentence so as to stress that our understanding of the 4.2 kyr event and its climate mechanisms in the data-rich heartland is considerably better than elsewhere in the world. Given the density of records in the region, this cannot be disputed. We also delete the last sentence of the paragraph which similarly stated "The global extent of the 4.2 kyr event is uncertain."

The new sentences read: "The climatic impact and mechanisms of the 4.2 kyr BP event are better understood in the data-rich heartland of Mediterranean Europe (Bini et al., 2019) and Middle East (Kaniewski et al., 2018) than elsewhere in the world, although even in the Mediterranean spatial heterogeneity limits a complete mechanistic understanding (Bini et al., 2019). While individual records do report climatic anomalies at 4.2kyr BP, the global picture remains incomplete. This is due to poorer spatial coverage of records, the use of low-resolution records with limited ability to reliably detect a 300-year anomaly, and chronological uncertainties inherent in paleoclimate records. These uncertainties hinder the determination of spatial variability and climate

mechanisms of the 4.2kyr event."

We believe it beyond the scope of this Indian Ocean focused paper to address and review every global record, and beyond the scope of a response to reviewers to comment on all 26 citations provided by the reviewer as potential expressions of the 4.2 kyr event. Other research groups have already taken up the task of systematic regional investigations elsewhere in the world and are likely to publish in the coming years.

FOURTH COMMENT: Our arguments concerning the Harappan relate to the seasonality of the drought, and do not infer any different timing of abandonment to any other previous study. We agree that there is ambiguity in the timing of abandonment between settlements, and in the paper we echo the idea that the precise timing of abandonment is due to a "complex interaction of societal, biogeophysical and geomorphological feedbacks and responses, rather than by climate alone". The archaeological evidence cannot simultaneously be sufficiently robust to allow for civilization collapse from a 4.2-3.9 kyr BP summer monsoon drought as reported widely in the literature, while at the same be insufficiently robust to not allow for civilization collapse from a 4.2-3.9 kyr winter monsoon drought proposed by our study. Professor Weiss argues simultaneously: "the archaeological evidence for the synchronous collapse of Egyptian, Mediterranean, West Asian, and Indus settlement systems at 4.2 ka BP appears increasingly robust." Yet also: "When, and at what rate, the Harappan urban abandonments occurred during this 700 year gross ceramic definition period is yet uncertain"

We believe this misunderstanding was caused by the phrase "The absence of a significant, widespread 4.2 kyr event in tropical Indian Ocean hydroclimate has consequences for the timing and causes of the deurbanization of the Harappan civilization in the Indus valley." As paleoclimatologists "tropical Indian Ocean hydroclimate" already infers summer, but we recognize that this might not be the case for other readers. We add the word summer to this sentence to clarify our point.

Our radiocarbon dates are taken from Sengupta et al., 2020 who state "Note the

near‐synchronous Harappan decline at all the sites just at the onset of or imme-
diately after the Meghalayan stage." All we did to the data was to sort the sites by
latitude. This reveals a clear geographical pattern which has been recognized previ-
ously in the literature (e.g. Giesche et al., 2019).

Our interpretation of the archaeological evidence for the timing of abandonment of
Harappan sites is no different to that of previous studies. Professor Weiss misrepre-
sents our interpretations by suggesting that we hypothesize widespread abandonment
at 4.2kyr BP. We hypothesize drought related climatic stress between 4.2 and 3.9 kyr
BP contributed to major city abandonment in the Indus Valley by 3.85 kyr BP (1900BC)
via complex biogeophysical and societal feedbacks, with ongoing aridity leading to
subsequent transition to a more rural society by 3.0 kyr BP. This hypothesis is already
widespread, and we suggest no changes to the timing of any societal change, merely
the seasonality of rainfall that contributed to it.

[Figure]

**Fig. 1.** Isotopic record of stalagmite KM-A from Mawmluh cave. The 4.2 kyr event is defined by the mid-point between the two excursions at 4.20 kyr BP (ie. between the two red circles)

[Figure]

Fig. 2. Hydroclimate records from East Africa northern hemisphere with at least seven data-point between 5000 and 3000 yr BP

---

## Author Comment (AC4) · 9 Feb 2021

We would like to thank Dr. Giesche, Professor Weiss and an anonymous reviewer for their inputs to this paper. The excellent questions have forced us to refine our paper and provide better evidence for our claims – be it on variability of the winter monsoon or a more thorough examination of the spatial relationships of the principal component analysis. We believe that the double drought hypothesis is far stronger with these inputs, our arguments have been made more robust with additional evidence, and that the manuscript is more precise and fairer in its language. In this final report we provide

a summary of the main discussion points raised by the reviewers, either individually or together. Individual detailed responses, and responses to minor comments have already been submitted as part of the discussion phase.

One of the major points raised by reviewer 1 was a desire to see more evidence for changes in the source region and transport route of the westerly winter monsoon. Our paper had originally focused on the summer monsoon, as this is where we provide new evidence. However, in order to establish the double drought hypothesis, we agree that both 'monsoons' need discussion. We are therefore happy to discuss rainfall changes along the path of westerly disturbances. Unfortunately, there are not sufficient high resolution, high dating precision records to conduct a similar PCA analysis as we have done for the Indian Ocean region. Therefore, our analysis was limited to a literature review. We first focused on records from the Middle East which give data about the winter season (e.g., a pollen record from Tell Tweini (Kaniewski et al., 2008), coral from the Gulf of Oman (Watanabe et al., 2019) and the speleothem d18O from Gol-E-Zard (Carolin et al., 2019)). These confirm the intuitive reasoning and previous studies (the Kaniewski et al., 2018 review in Climate of the Past) that the 4.2-3.9 kyr BP drought was likely a winter drought in the Middle East. We then tracked the drought through non-seasonally resolved records across the Middle East towards the Indus Valley. While not every record showed a drought (e.g., Mirabad Lake, (Stevens et al., 2006), Mahar-lou Lake (Djamali et al., 2009)) (it would be quite unusual if all records did agree), a substantial number of lake records did show a drying from 4.2-3.9 kyr BP (e.g. Neor lake (Sharifi et al., 2015), Lake Hamoun, (Hamzeh et al., 2016). This also helps put into context the archaeobotanical data determined from Harappan crops - the carbon isotopes of rice grains, a summer crop, show drought only after 4.0kyr BP in both Gujarat and Indus valley locations (Kaushal et al., 2019). Overall, we have made a substantial addition to Section 4.3 (Regional Climate Drivers) to explain this, and feel the double drought hypothesis is strengthened and the paper substantially improved by these changes.

The suggestion of a winter drought (and therefore two droughts) is not new. As Dr. Giesche pointed out, the idea was raised previously in their 2019 Climate of the Past paper. It was our error not to include it. Our analysis oversimplified the relationship between the individual proxies of their paper (G. ruber and N. dutertrei) as summer and winter rainfall proxies, whereas it was the relationship between the two (and G. sacculifer) that was interpreted as the winter rainfall proxy. We are happy to correct our error and give credit where it is due, making changes in several places in the manuscript. We feel there are plenty of similarities between the two records, and they are not necessarily at odds with each other: e.g. a secular trend of decreasing summer rainfall, an increase in winter rainfall at 4.6-4.5 kyr BP, a winter rainfall drought around the 4.2 kyr event, an increase in rainfall between 3.5 and 3.0 kyr BP are all consistent. There remain some differences between the two records. Giesche et al., hypothesise concurrent winter and summer droughts. We hypothesize consecutive winter then summer droughts. We believe this difference might be due to the Giesche et al winter and summer rainfall interpretations both relying on the G. ruber proxy (although the difference between N dutertrei and G. sacculifer support their G.ruber based interpretations). They therefore might not be truly independent records.

Reviewer 1 also asked us to provide further discussion on the spatial component of the MC-PCA, particularly PC2. We agree that the spatial analysis was not discussed as thoroughly as it might. The spatial pattern of loadings, particularly PCA-3 PC2, help provide key evidence in our discussion as to why the 4.2-3.9 kyr drought is unlikely to be caused by changes in summer rainfall. Namely that both the Oman and Sahiya cave records plot oppositely to the Indus fan records. We are happy to follow the advice of reviewer two and include maps (a new figure 5) and an enhanced discussion of the spatial characteristics of PC1 and PC2 (second paragraph of section 4.3).

Reviewer one asked us to provide technical points on the Monte-Carlo procedure used and to share the data and code from this work:

Re: technical issues with code: We have used the Deininger et al. (Climate Dynamics 2017) code, with only minor adjustments in consultation with Michael Deininger to facilitate changes to the input files (the time period was previously hard coded at 30 years) and output files for easier plotting. As raised by Reviewer 1, truncation of the records is part of the Deininger software. The production of 1000 independent age models from each record results in individual realisations being stretched and compressed between randomly sampled ages. Windows without data-points are dropped. Therefore, there is a compromise between window size and the ability to generate one data point in each and every time window for all 10,000 age models (10 records x 1000 realisations). Slightly longer windows did not result in substantially longer principal component time-series.

Re: data availability. We thoroughly agree with the idea that data should be shared. We are submitting both the new Madagascar record (in companion paper cp-2020-137) to SISAL and the NOAA paleoclimatology database, and will submit the principal component time series and loadings to the NOAA paleoclimatology database also. The majority of records used in this study are publicly available. However, we do not believe it is our place to share the data of others so will refrain from publishing their data for all the records used in this study.

Re: code availability. We feel similarly about the Deininger MC-PCA code used in this study. It is not our place to share other people's code without their express permission. Reviewer one asks for both code to be available (to facilitate replication) but also changes to the code (to facilitate non-independent age models in the Indus fan sediment cores). These are unfortunately incompatible. We have decided to stick as closely as possible to the Deininger code, for the sake of replication, rather than adjust the code for modest gains in performance and make our code unreplicable as we cannot republish what is 95% someone else's work.

Professor Weiss detailed four significant comments in response to our manuscript. The comments, and our detailed response total over 7600 words. Here we provide a summary of the main issues:

The first comment took issue with our interpretation of KM-A stalagmite from Mawmluh Cave, which Professor Weiss argues is an Indian Summer Monsoon record only. A recent drip monitoring study by Ronay et al. (Scientific Reports 2019) casts doubt on this idea, instead suggesting that stalagmite proxies (both stable isotopes and trace metals) may have a strong or even dominant winter component. There is therefore doubt as to whether KM-A is a 100% Indian Summer Monsoon record. The KM-A record is still useful and under the interpretation of the double drought hypothesis, does show convincing evidence for all three drying events. We would therefore argue that KM-A likely records both summer and winter rainfall variability.

Professor Weiss then argues that the KM-A record replicates with stalagmites ML.1 and ML.2 "within standard deviations". We disagree. While ML.1 nor ML.2 may contain minor changes to variability or mean state around the 4.2 kyr event, neither contain substantial step-change deviations of the kind seen in KM-A at either 4.3 or 4.1 kyr BP. Certainly neither ML.1 nor ML.2 contains the dramatic reversal at 3.9 kyr BP seen in KM-A. ML.1 and KM-A are plotted together in figure 2 of the manuscript. This is followed by an objection to one of our final statements that KM-A is unsuitable as a golden spike for the mid- to late- Holocene transition. We retract our comment that the record is of low resolution, it is not. But we stand by our comments that the stalagmite has low dating frequency, is not replicable within its own cave, ambiguously defines a climate event, and is not significant across its climate domain (i.e. the Indian Summer Monsoon). In our direct response we provide detailed evidence for our stance on each of these clauses.

The second comment takes issue with the records used in the Principal Component Analysis. Our entire analysis was designed to investigate only the highest resolution records available so that the common centennial scale variability could be determined taking into account age model uncertainty. Including low-resolution and poorly dated records would not provide statistically significant insight into the 4.2 kyr event given the age uncertainty. The records chosen for our PCA were chosen on the basis of

resolution and length of record and were selected objectively based on these criteria. Professor Weiss criticized a lack of discussion on global low-resolution records generally, and East African records in particular. In our manuscript we chose to discuss low resolution records in two sub-regions only: the south-west Indian Ocean in the companion manuscript cp-2020-137, and the Indian subcontinent in this manuscript. The East African records were not included as we felt that a discussion of the impacts of the 4.2kyr event on the East African double wet season (not part of the Indian Summer Monsoon domain) was beyond the scope of this Indian Summer Monsoon and Indian Winter Monsoon paper. Nevertheless, we conducted a literature review of low resolution East African records and could find no compelling evidence of a 4.2 kyr event in summer rainfall that could cast significant doubt on our basin wide analysis (full details of that review are provided in the response). On a similar note, the impact of the 4.2kyr event on the collapse of the Old Kingdom in Egypt was out of scope.

The third comment questioned the phrase "the areal extent of the 4.2 kyr BP beyond the data-rich heartland of Mediterranean Europe (Bini et al 2019) and Mesopotamia (Kaniewski et al 2018) is unclear". We agree that this phrase is likely an overstatement, and as the discussion of every single global record covering the 4.2kyr BP event is beyond the scope of this paper, we are happy to rephrase this statement. Instead, we highlight that while numerous climatic anomalies at the mid- to late- Holocene transition have been attributed to the 4.2kyr event, the global picture remains insufficient to attribute climate mechanisms.

The fourth comment discusses the radiocarbon data from the Indus Valley. Professor Weiss argues simultaneously that: "When, and at what rate, the Harappan urban abandonments occurred during this 700 year gross ceramic definition period is yet uncertain" While arguing later in his response that: "the archaeological evidence for the synchronous collapse of Egyptian, Mediterranean, West Asian, and Indus settlement systems at 4.2 ka BP appears increasingly robust." These two statements are contradictory. Moreover, we make no changes to the interpretation of the radiocarbon data.

[Figure]

The radiocarbon data compilation used in our study is taken from the Sengupta et al., 2020 compilation which argues for 4.2-3.9kyr collapse at Dholivara. The idea that abandonment was most severe in the Indus Valley, and did not occur in simultaneously in Gujurat is not a new idea either (e.g. Giesche et al., 2019). The archaeological evidence cannot simultaneously be sufficiently robust to allow for civilization collapse from a 4.2-3.9 kyr BP summer monsoon drought, while at the same be insufficiently robust to allow for civilization collapse from a 4.2-3.9 kyr winter monsoon drought. Our data does not call into question a 4.2-3.9 kyr decline in the Harappan civilization and abandonment of the more northerly Indus valley sites. Our data provides evidence that a winter drought occurred at this time, and not a summer drought. We agree that there is remaining uncertainty in the radiocarbon ages, both in their inherent age uncertainty, but also in the archaeological context in which they are found. However, if the uncertainties are sufficiently concerning that the chronology of civilization decline and collapse cannot be attributed to the 4.2 kyr event, then there are dozens of previously published studies across numerous disciplines that are unsubstantiated.

Our manuscript provides perhaps the first compelling climate mechanism to extend the influence of the 4.2 kyr event outside of the Middle East and Mediterranean (both climatically and culturally). We argue that the 4.2kyr event that contributed to the collapse of the Akkadian Empire in the Middle East is likely to influence the Harappan by the same mechanism: reduced moisture from the Mediterranean transported by upper-level winter storms in the westerly jet. This data fits well with the regional climate dynamics – the 4.2kyr event is a known winter drought in the Middle East, and the Middle East is the moisture source and transport region for the westerly disturbances that feed the Indian Winter Monsoon. A winter drought is in agreement with the archaeobotanical data and the temporal pattern of settlement abandonment. Our analysis expands, rather than limits, the reach of 4.2 kyr event.

Finally, In our final report, the editor asked us to comment on the recently published Lilaur Lake record from the Ganga Basin (Singh et al., 2021). Singh et al interpret

coarse silt and clay fractions as wet/dry indicators with increased coarser fractions representing wetter conditions. Under this interpretation they infer a dry event from 4.20 (possibly 4.25) to 4.05 kyr BP, wet or very wet conditions from 4.05 to 3.8 kyr BP and then a gradual drying towards 3.0kyr BP. Therefore, at first viewing, it appears to match the PC1 pattern of our analysis but with the opposite loading to the other records.

The Lilaur Lake record is good quality and has excellent sampling resolution. However, the dating of this section of core is not ideal. There is one radiocarbon date at 3.1 kyr BP +-35 and one OSL date of 3.7 kyr BP +-300 covering the entire Holocene. The 4.2 kyr event is interpolated between the 3.7 kyr BP age and a radiocarbon date at 17.4 kyr BP. Therefore, the precise location of a 200 year long dry interval is difficult to interpret.

Despite this uncertainty in precision, it is still possible to interpret the record in light of our results under the assumption the mean age model is accurate. As suggested by Singh et al., 2021, the Lilaur Lake record does look remarkably similar to the Tso Moriri Lake EM3 record of Dutt et al. (2018), which has more precise bracketing dates at 4.4kyr BP +-120yr and 3.7kyr BP +-50yr. The Tso Moriri element record shows an abrupt drying around 4.35 kyr BP but it is unidirectional in nature, lasting until at least 3.4 kyr BP. Tso Moriri Lake has significant winter rainfall under modern climatology. It is therefore reasonably intuitive to suggest that the Tso Moriri Lake likely experienced consecutive winter and then summer droughts. It is a little harder to interpret the Lilaur Lake record in this way, as the modern climatology at Lilaur Lake is dominated by the summer monsoon. However, if the Giosan et al. (2018) hypothesis (supported by Giesche et al., 2019 and our analysis) of increased westerly disturbance rainfall is correct, then Lilaur Lake is not too far east that winter rainfall may have penetrated as far as Lilaur Lake and influenced proxy variability. The sediment core itself is a wet/dry indicator with no seasonality indicators.

What about the wet event from 4.05 to 3.80kyr BP? A coring gap (not a hiatus) between 4.7 and 4.25 kyr BP makes it difficult to evaluate the 4.2 kyr event in the context of the

prior climatic regime, particularly the 4.6-4.3 kyr BP wet period (Giosan et al., 2018, Giesche et al., 2019, this study). Therefore, it is impossible to know if the 4.6-4.3 kyr period was wetter than the 3.9-3.6 kyr BP period. The coarse silt percentage indicates that 4.05 to 3.80kyr BP was the wettest period on record. However, the clay percentage suggests that 4.05 to 3.80kyr BP was wet, but not unusually so – being comparable or perhaps slightly drier than 5.1 to 4.8 kyr BP. (Prior to 5.1 kyr BP the lake was likely a river so direct comparison is more difficult). The strength of the 4.05 to 3.80kyr BP wet signal at Lilaur Lake is unclear.

Our discussion of the Lilaur Lake record is emblematic of many hydrological records in the region, and could be repeated for any and all Indian subcontinent hydrocli-mate record. Most of the records are interpreted entirely as Indian Summer Monsoon records, as the modern climatology is dominated by summer rainfall. If Giosan, Gi-esche and ourselves are correct in interpreting a stronger winter monsoon at the end of the mid-Holocene then it is entirely possible that many paleoclimate records were recorders of winter rainfall variability or mixed winter and summer (i.e. annual) rainfall variability. The Double Drought hypothesis provides a new framework with which to view anomalies and trends in the Indian Ocean hydroclimate system of the mid to late Holocene, made up of three major components either in isolation or as a mixture. They are: 1) gradual drying over millennia, 2) abrupt drying at 4.25kyr BP with recovery at 3.9 kyr BP and 3) abrupt drying at 4.0 kyr BP. Some records will fit more naturally than others, and there will be differences between different proxies from the same location depending on proxy sensitivity. In the case of Tso Moriri the Dutt et al., 2018 elemental record supports abrupt drying at 4.3 kyr BP and no recovery until 3.4 kyr BP, which fits nicely with both #2 and #3 – recording winter and then summer drought. The Leipe et al. (2014) pollen record shows both 4.4 kyr BP and 4.0 kyr BP drying, interpretable as #2 and #3, but with a slight recovery in between. In the case of Lilaur Lake, the clay per-centage record shows a gradual drying trend, decrease at 4.2 kyr BP and only a slight recovery from 3.9 kyr BP onwards: interpretable as showing #1 and #2 clearly, but #3 less obviously but plausible. The Lilaur Lake record coarse silt percentage shows a

gradual drying trend, decrease at 4.2 kyr BP but no drying at 3.9 kyr BP, indeed a full recovery to wet conditions: #1, #2, but definitely not #3. Such similarities but subtle differences between records are a feature of all records which measure more than one hydrological proxy in the same archive. Spatial heterogeneity will still exist in the climate system and uncertainty will still exist in the interpretation of hydrological proxies and their sensitivity to the varying and variable components of the hydrological cycle (winter rainfall, summer rainfall, annual rainfall, evaporation, storminess, river flooding etc.). There is still much to learn.

Singh, S., Gupta, A.K., Rawat, S., Bhaumik, A.K., Kumar, P., Raj, S.K.: Paleomonsoonal shifts during âĹij13700 to 3100 yr BP in the central Ganga Basin, India with a severe arid phase atâĹij4.2 ka, Quaternary International, in press, https://doi.org/10.1016/j.quaint.2021.01.015.